# Cerebral mGluR5 availability contributes to elevated sleep need and behavioral adjustment after sleep deprivation

Sebastian C Holst[1,2†], Alexandra Sousek[1,3†], Katharina Hefti[1], Sohrab Saberi-Moghadam[3], Alfred Buck[4], Simon M Ametamey[5,6,7], Milan Scheidegger[8,9], Paul Franken[3], Anke Henning[5,6,7], Erich Seifritz[2,8], Mehdi Tafti[3,10]*, Hans-Peter Landolt[1,2]*

[1]Institute of Pharmacology and Toxicology, University of Zürich, Zürich, Switzerland; [2]CRPP Sleep and Health, Zürich Center for Interdisciplinary Sleep Research, University of Zürich, Zürich, Switzerland; [3]Center for Integrative Genomics, University of Lausanne, Lausanne, Switzerland; [4]Division of Nuclear Medicine, University Hospital Zürich, Zürich, Switzerland; [5]Center for Radiopharmaceutical Sciences of ETH, Zürich, Switzerland; [6]Paul Scherrer Institut, Zürich, Switzerland; [7]University Hospital of Zürich, Zürich, Switzerland; [8]Department of Psychiatry, Psychotherapy and Psychosomatics, Psychiatric Hospital, University of Zürich, Zürich, Switzerland; [9]Institute for Biomedical Engineering, University of Zürich and ETH Zürich, Zürich, Switzerland; [10]Department of Physiology, Faculty of Biology and Medicine, University of Lausanne, Lausanne, Switzerland

*For correspondence:
mehdi.tafti@unil.ch (MT);
landolt@pharma.uzh.ch (H-PL)

†These authors contributed equally to this work

Competing interests: The authors declare that no competing interests exist.

**Abstract** Increased sleep time and intensity quantified as low-frequency brain electrical activity after sleep loss demonstrate that sleep need is homeostatically regulated, yet the underlying molecular mechanisms remain elusive. We here demonstrate that metabotropic glutamate receptors of subtype 5 (mGluR5) contribute to the molecular machinery governing sleep-wake homeostasis. Using positron emission tomography, magnetic resonance spectroscopy, and electroencephalography in humans, we find that increased mGluR5 availability after sleep loss tightly correlates with behavioral and electroencephalographic biomarkers of elevated sleep need. These changes are associated with altered cortical myo-inositol and glycine levels, suggesting sleep loss-induced modifications downstream of mGluR5 signaling. Knock-out mice without functional mGluR5 exhibit severe dysregulation of sleep-wake homeostasis, including lack of recovery sleep and impaired behavioral adjustment to a novel task after sleep deprivation. The data suggest that mGluR5 contribute to the brain's coping mechanisms with sleep deprivation and point to a novel target to improve disturbed wakefulness and sleep.
DOI: https://doi.org/10.7554/eLife.28751.001

## Introduction

'Why do we sleep?' is one of the remaining unanswered questions in biomedical research. To elucidate function(s) of sleep, the molecular substrate(s) of sleep homeostasis, that is, increased sleep need and intensity after prolonged wakefulness (*Achermann and Borbély, 2011*), need to be identified. Dynamic changes in electroencephalographic (EEG) slow-wave (SWA) or delta activity (0.5–4.5 Hz), and the slow oscillation (<1 Hz) in non-rapid-eye-movement (NREM) sleep, reliably index sleep need. These fundamental sleep rhythms are thought to reflect neural plasticity across the sleep-wake cycle (*Krueger et al., 2013*; *Tononi and Cirelli, 2014*). They are generated in the intact brain

by complex cortical networks, homeostatically controlled, and provide neurophysiological markers for sleep-dependent memory consolidation and waking plasticity determining sleep need (*Crunelli and Hughes, 2010*; *Hung et al., 2013*; *Marshall et al., 2006*; *Matsuzaki et al., 2004*). The strong genetic control of these neurophysiological markers of sleep homeostasis (*Franken et al., 2001*; *Landolt, 2011*) demonstrates that sleep need has a molecular substrate.

The activity-dependent expression in cortex and hippocampus of the immediate early gene Homer1a currently provides the most specific molecular correlate of sleep-deprivation-induced rebound in delta power in mice (*Maret et al., 2007*). In humans, sleep slow waves systematically recruit cortical and subcortical brain regions with high availability of metabotropic glutamate receptors of subtype 5 (mGluR5), including medial superior frontal cortex, dorso-lateral prefrontal cortex, inferior parietal cortex, and precuneus (*Ametamey et al., 2007*; *Dang-Vu et al., 2008*; *Hefti et al., 2013*). Homer1a selectively uncouples mGluR5 from effector targets in the post-synaptic density (*Kammermeier and Worley, 2007*). This uncoupling attenuates mGluR5-mediated activation of phospholipase C, production of inositol trisphosphate ($IP_3$) and diacyl glycerol, rise in intracellular calcium, and downstream signaling (*Berridge, 2016*). The interaction between Homer1a and mGluR5 can induce synaptic long-term depression (LTD) (*Ronesi and Huber, 2008*) and ensure stable neuronal excitability despite local changes in synaptic inputs (*Hu et al., 2010*). Such forms of neural plasticity contribute to the fine-tuning of synaptic strength across the sleep-wake cycle (*Diering et al., 2017*; *Krueger et al., 2013*). Consistent with this view, sleep deprivation increases availability of mGluRs in the human and rat brain (*Hefti et al., 2013*; *Tadavarty et al., 2011*). Furthermore, specific mGluR5 negative allosteric modulators promote and consolidate sleep in rats, whereas positive allosteric modulators promote wakefulness (*Ahnaou et al., 2015a*). Based upon this background, the current work tackled the next essential steps in defining the exact roles of mGluR5 in physiological sleep-wake regulation. Thus, an unparalleled multimodal approach was applied along a translational axis in humans and mice that combined behavioral, molecular brain imaging and neurophysiological studies before and after sleep deprivation, to investigate the involvement of mGluR5 in regulating sleep need.

## Results

### mGluR5 availability predicts behavioral and EEG markers of sleep need in humans

Twenty-six healthy male volunteers were kept awake for 40 hr under constantly supervised, controlled conditions. Using positron emission tomography (PET) with the highly selective, non-competitive radio-ligand, [11]C-ABP688 (*Ametamey et al., 2006*; *Ametamey et al., 2007*), mGluR5 receptor availability was quantified twice, after ~9 (baseline: at 16:39 ± 8 min) and ~33 hr (sleep deprivation: at 16:30 ± 7 min) of wakefulness, in randomized, cross-over order. PET imaging sessions lasted for 67 ± 0.6 min and were immediately followed by ~45 min magnetic resonance spectroscopy ([1]H-MRS) imaging, to investigate mGluR5-associated metabolic changes (see *Figure 1A* for study-protocol). During PET and [1]H-MRS imaging, wakefulness was verified by polysomnography (during PET imaging) or continuous button press on a hand-held response box (during [1]H-MRS imaging).

Apart from a global increase in mGluR5 binding ($DV_{norm}$: 1.51 ± 0.02 *vs.* 1.56 ± 0.02, p<0.004), which was more pronounced in individuals with lower baseline mGluR5 availability than in individuals with higher mGluR5 availability (*Figure 2—figure supplement 1*), prolonged wakefulness also had behavioral consequences. Although subjects were alerted over the intercom when polysomnographic signs of sleep occurred during PET scanning, unintended sleep could not be completely prevented. Stage 1 NREM sleep was more than three times more prevalent after sleep deprivation than after normal sleep (14.0 ± 3.1 *vs.* 4.0 ± 1.8%; $t_{25}$ = 4.50, p<0.0001). When comparing the increase in stage one sleep episodes with the increase in global mGluR5 availability, a significant positive association emerged (*Figure 2A*).

To estimate homeostatic sleep need at sleep onset, EEG delta (0.5–4.5 Hz) and <1 Hz activity were quantified in the first NREM sleep episodes of all-night polysomnographic recordings in baseline and recovery nights (*Iber et al., 2007*). In both conditions, global mGluR5 availability was correlated with initial delta (*Figure 2C and D*), as well as <1 Hz activity ($r_S$ (Baseline) >0.71, $r_S$ (Recovery) >0.71, $p_{all}$ <0.0002, n = 23) (*Figure 2—figure supplement 2*). Intriguingly, local mGluR5

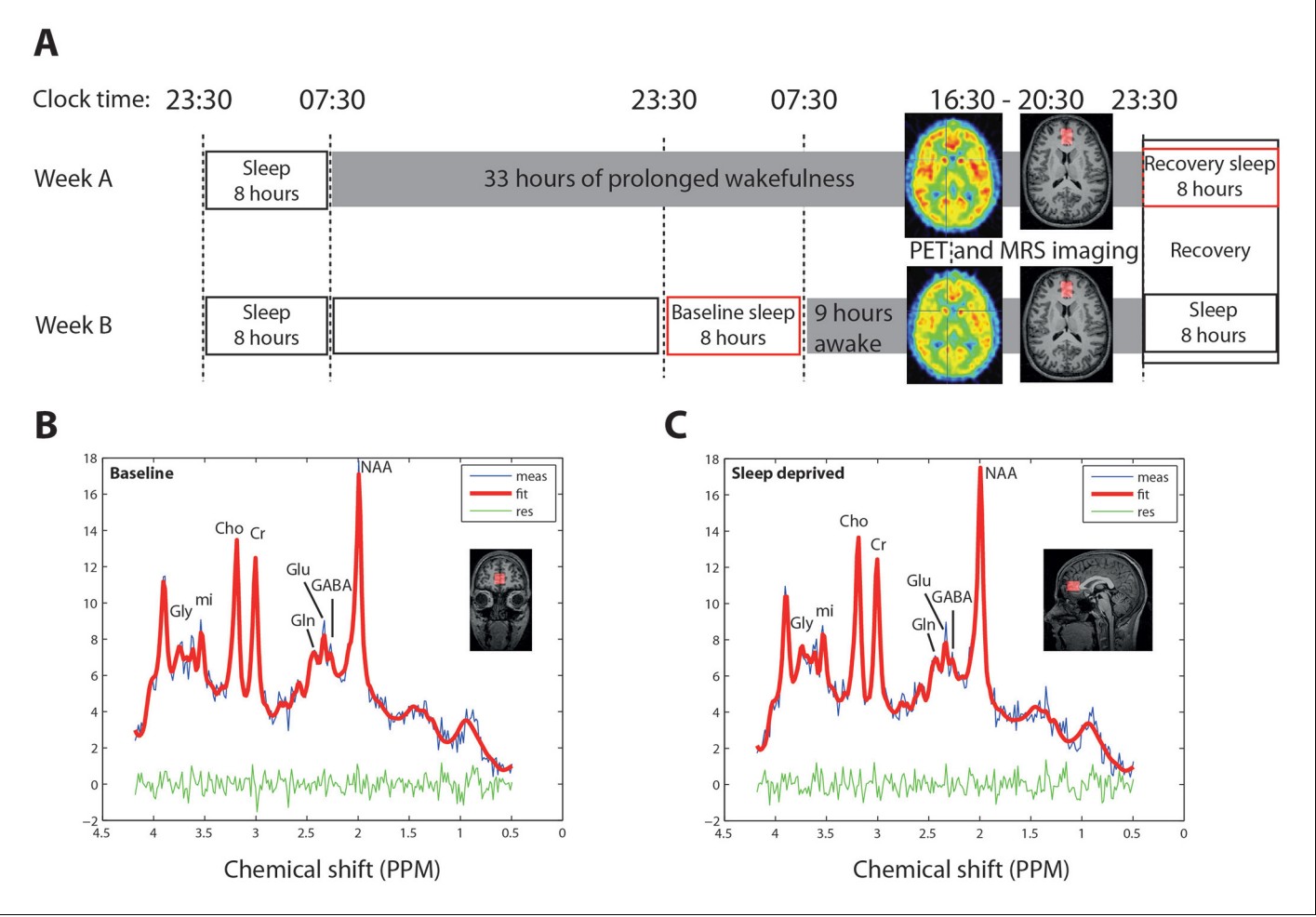

**Figure 1.** Sleep deprivation and imaging protocol and representative [1]H-MR spectroscopy. (**A**) Randomized, cross over study design, with PET imaging after 9 and 33 hr of wakefulness. MR-spectroscopy was performed directly after PET, with a delay of 45–100 min. Polysomnographic recordings of 8 hr baseline and recovery nights are illustrated by red boxes. (**B–C**) Prototypical projections in one representative healthy individual of two-dimensional J-resolved single-voxel [1]H-magnetic resonance spectroscopy spectra acquired in the bilateral pregenual anterior cingulate cortex (pgACC; red overlay) in sleep control (**B**) and sleep deprived (**C**) condition. Red lines represent the Profit 2.0 model fit, blue lines raw data, and the green lines the residuals (error). Gly: Glycine, mi: myo-inositol, Cho: choline, Cr: total creatine and phosphocreatine, Gln: glutamine, Glu: glutamate, GABA: gamma-Aminobutyric acid, NAA: N-acetylaspartate.
DOI: https://doi.org/10.7554/eLife.28751.002

availability in medial superior frontal cortex, dorso-lateral prefrontal cortex, inferior parietal cortex, and precuneus explained roughly half of the inter-individual variation in <1 Hz activity in the first NREM sleep episode in baseline and recovery sleep (*Figure 3* and *Table 1*). Thus, those individuals with the most pronounced mGluR5 binding in this neuroanatomic network regulating the sleep slow oscillation (*Dang-Vu et al., 2008*) exhibited the highest values in this marker of homeostatic sleep need. A similar relationship was also observed for the entire 0.5–4.5 Hz band (yet not any other frequency band), although it did not withstand the stringent Bonferroni correction ($p<0.00278$; *Figure 3—figure supplement 1*).

Consistent with the concept of sleep homeostasis (*Achermann and Borbély, 2011*), sleep deprivation strongly increased delta ($47.9 \pm 7.2\%$) and <1 Hz activity ($38.5 \pm 6.7\%$; $p_{all} < 0.0001$) compared to baseline. Intriguingly, subdividing the study participants by median split into groups with low and high change in global mGluR5 availability after sleep deprivation revealed that the group with a minor change in mGluR5 availability also showed a reduced increase in delta and <1 Hz activity when compared to the group with a more pronounced increase in mGluR5 (*Figure 2—figure*

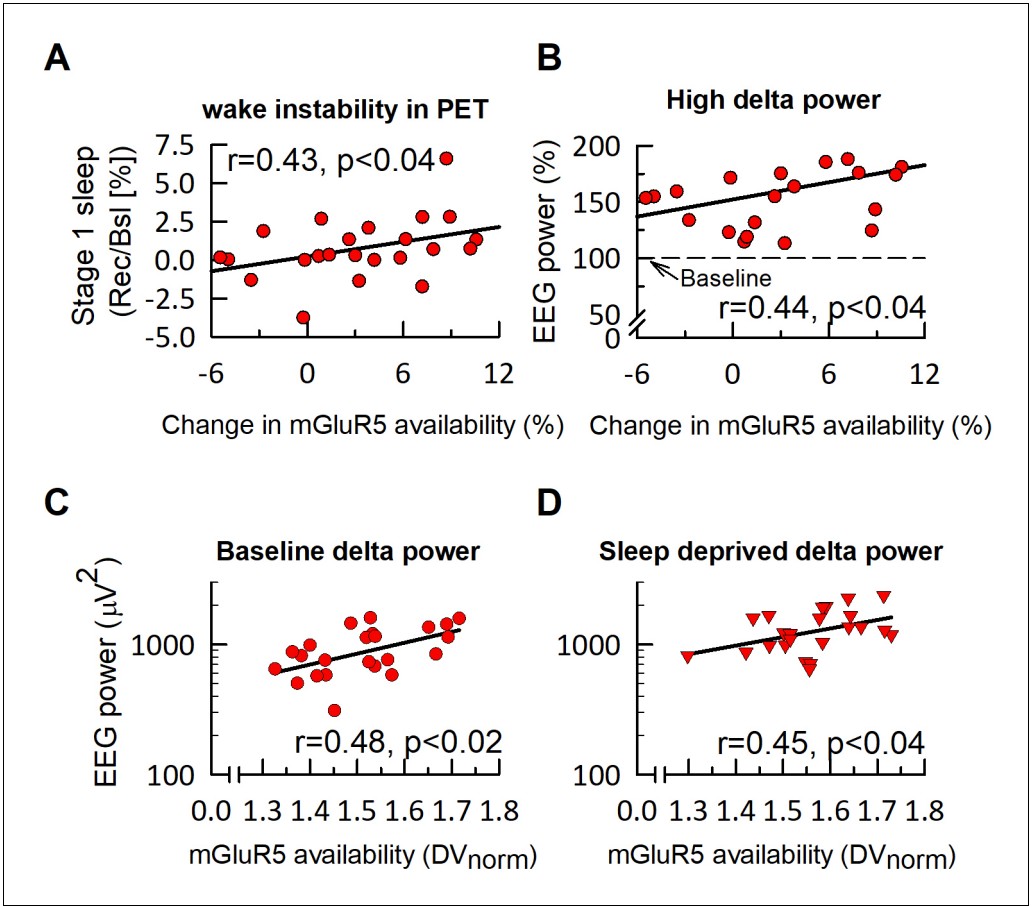

**Figure 2.** mGluR5 availability in humans is associated with EEG delta activity in NREM sleep. The relative change in global mGluR5 availability after sleep deprivation is significantly associated with the increase in intermittent stage one sleep ('wake instability') during PET imaging (**A**) and 2–4 Hz ('high delta') activity in subsequent NREM sleep in the recovery night (**B**). In both, baseline (**C**) and sleep deprivation (**D**) conditions, EEG delta activity (0.5–4.5 Hz, log scale) in the first NREM episode is positively associated with global mGluR5 availability. Statistics indicate Spearman rank correlation coefficients (r).
DOI: https://doi.org/10.7554/eLife.28751.003

The following figure supplements are available for figure 2:

**Figure supplement 1.** Individual change in mGluR5 availability as a function of baseline mGluR5 expression.
DOI: https://doi.org/10.7554/eLife.28751.004

**Figure supplement 2.** Spearman rank correlation coefficients between global mGluR5 availability and EEG power between 0 and 20 Hz in baseline and sleep deprivation conditions, and the change caused by sleep loss.
DOI: https://doi.org/10.7554/eLife.28751.005

**Figure supplement 3.** Enhanced rebound after sleep deprivation in EEG delta (0.5–4.5 Hz) and <1 Hz activity in individuals with high change in global mGluR5 availability after sleep deprivation.
DOI: https://doi.org/10.7554/eLife.28751.006

*supplement 3*). When comparing the waking-induced increase in EEG power between 0.25 and 20 Hz and the increase in global mGluR5 availability, significant correlations within the delta band were found. Mean power in the 2–4 Hz ($r_S$ = 0.44, p<0.04) (*Figure 2B*) and <1 Hz ranges ($r_S$ = 0.39, p<0.07) were specifically associated with increased mGluR5 availability (*Figure 2—figure supplement 2*). These results provide the first puzzle piece of evidence for the hypothesis that functional mGluR5 availability not only correlates with absolute low-frequency EEG power, but represents a molecular marker of elevated sleep need in response to sleep loss in humans.

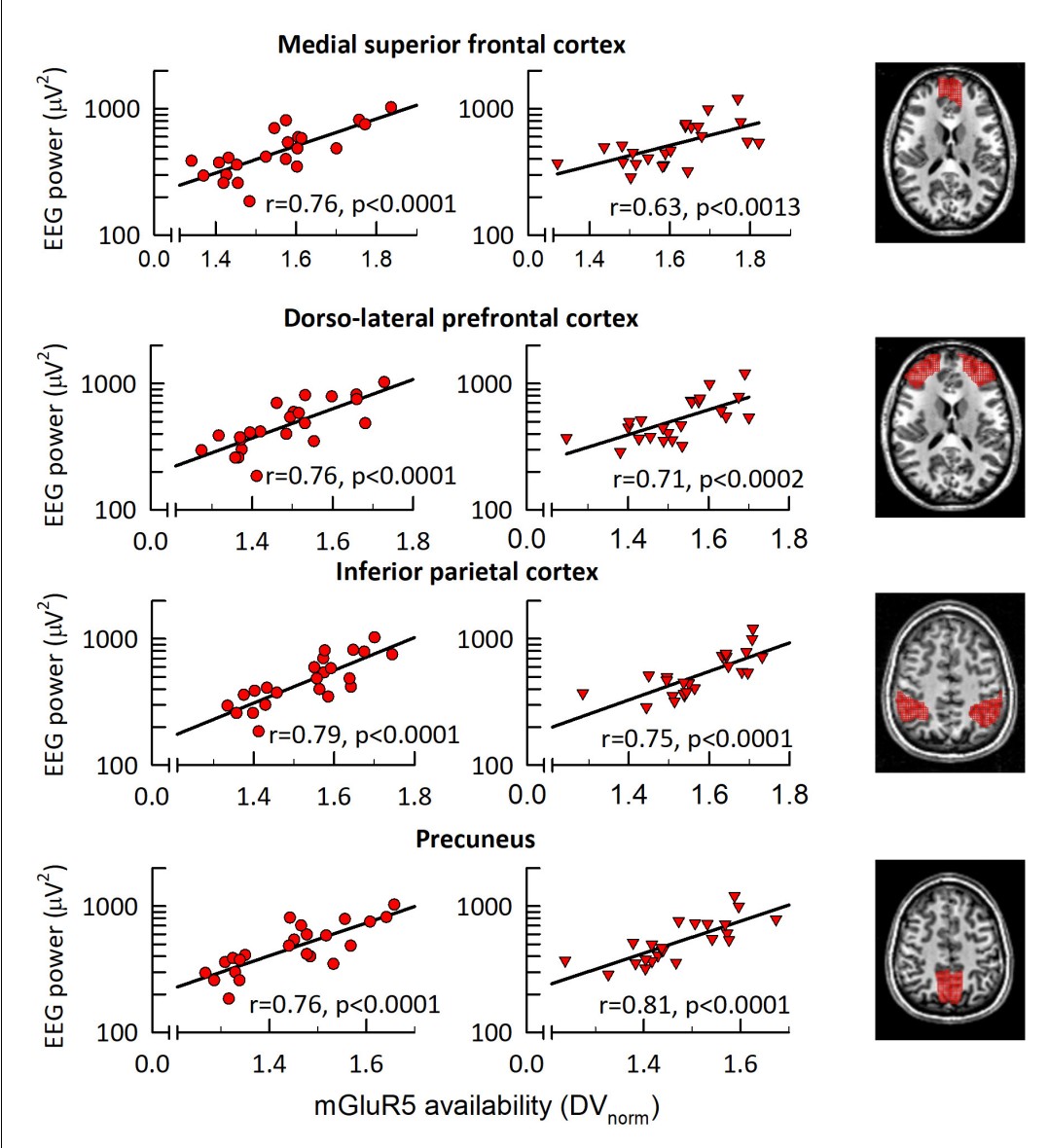

**Figure 3.** mGluR5 availability in fronto-parietal network correlates with EEG <1 Hz activity in NREM sleep. Association between the EEG slow oscillation activity (0.25–1.0 Hz) in the first NREM sleep episode of sleep control (left) and sleep deprivation (middle) conditions. Correlation plots: Significant ($p_{all} < 0.0015$) Spearman rank correlation coefficients (r) between absolute EEG <1 Hz activity (C3M2 derivation) and regional mGluR5 availability in medial superior frontal cortex, dorso-lateral-prefrontal cortex, inferior parietal cortex, and precuneus. Right column: Axial slices illustrating the brain regions (red overlay) showing a significant association between EEG <1 Hz activity and mGluR5 availability. Z-coordinates according to the MNI brain atlas: medial superior frontal cortex, z = 16; dorso-lateral-prefrontal cortex, z = 16; inferior parietal cortex, z = 50; precuneus, z = 48.

DOI: https://doi.org/10.7554/eLife.28751.007

The following figure supplement is available for figure 3:

**Figure supplement 1.** Correlations between EEG power bands and regional mGluR5 availability.

DOI: https://doi.org/10.7554/eLife.28751.008

## Sleep deprivation-induced changes in brain metabolites may reflect downstream markers of mGluR5 activation

To examine whether the strong association between mGluR5 availability and behavioral and neuro-physiological markers of sleep need may be linked to endogenous changes in brain chemistry, 17 study participants underwent [1]H-MRS imaging immediately following PET scanning. A previously established [1]H-MRS protocol of a single voxel within the pregenual anterior cingulate gyrus of the

**Table 1.** Correlation between mGluR5 availability in distinct brain regions and the EEG <1 Hz activity.

| | | Correlation between mGluR5 availability and EEG < 1 Hz activity | | | |
| | | Baseline | | Sleep deprivation | |
| Brain region | | $r_P$, p | $r_S$, p | $r_P$, p | $r_S$, p |
|---|---|---|---|---|---|
| Medial superior frontal cortex * | Left | 0.75, <0.0001 | 0.73, <0.0001 | 0.60, 0.0022 | 0.63, 0.0011 |
| | Right | 0.79, <0.0001 | 0.76, <0.0001 | 0.60, 0.0024 | 0.66, 0.0006 |
| Orbitofrontal cortex | Left | 0.60, 0.0027 | 0.60, 0.0026 | 0.58, 0.0036 | 0.57, 0.0046 |
| | Right | 0.68, 0.0003 | 0.69, 0.0002 | 0.57, 0.0046 | 0.56, 0.0051 |
| Dorsolateral prefrontal cortex * | Left | 0.76, <0.0001 | 0.77, <0.0001 | 0.67, 0.0005 | 0.66, 0.0006 |
| | Right | 0.75, 0.0001 | 0.76, <0.0001 | 0.67, 0.0004 | 0.72, 0.0001 |
| Anterior cingulate cortex | Left | 0.67, 0.0004 | 0.70, 0.0002 | 0.51, 0.0126 | 0.59, 0.0032 |
| | Right | 0.66, 0.0007 | 0.64, 0.001 | 0.55, 0.0061 | 0.61, 0.0021 |
| Inferior parietal cortex * | Left | 0.80, <0.0001 | 0.79, <0.0001 | 0.77, <0.0001 | 0.76, <0.0001 |
| | Right | 0.79, <0.0001 | 0.78, <0.0001 | 0.71, 0.0002 | 0.75, <0.0001 |
| Precuneus * | Left | 0.82, <0.0001 | 0.77, <0.0001 | 0.78, <0.0001 | 0.80, <0.0001 |
| | Right | 0.77, <0.0001 | 0.76, <0.0001 | 0.76, <0.0001 | 0.82, <0.0001 |
| Insula | Left | 0.63, 0.0014 | 0.63, 0.0012 | 0.47, 0.0244 | 0.41, 0.0532 |
| | Right | 0.69, 0.0003 | 0.67, 0.0004 | 0.58, 0.0041 | 0.58, 0.004 |
| Striatum | Left | 0.62, 0.0016 | 0.63, 0.0013 | 0.50, 0.0147 | 0.49, 0.0176 |
| | Right | 0.68, 0.0004 | 0.66, 0.0006 | 0.52, 0.0104 | 0.58, 0.0037 |
| Parahippocampal gyrus | Left | 0.41, 0.0536 | 0.46, 0.0267 | 0.33, 0.1296 | 0.28, 0.1897 |
| | Right | 0.53, 0.0093 | 0.53, 0.01 | 0.38, 0.0728 | 0.40, 0.0585 |
| Hippocampus | Left | 0.52, 0.0101 | 0.47, 0.0245 | 0.36, 0.0886 | 0.37, 0.0853 |
| | Right | 0.53, 0.009 | 0.48, 0.0198 | 0.43, 0.0413 | 0.37, 0.0835 |

$r_P$ = Pearson Product Moment Correlation coefficient; $r_S$ = Spearman Rank Correlation coefficient. Those brain regions that showed a significant ($p_{corr}$ <0.00278) correlation between mGluR5 availability and EEG <1 Hz activity on left and right hemisphere in both baseline and sleep deprivation conditions are highlighted by a star (*).
DOI: https://doi.org/10.7554/eLife.28751.009

medial prefrontal cortex was employed (*Hulka et al., 2016*). Changes in metabolite concentrations were considered relevant if simultaneously fulfilling the following criteria: (1) significant alteration by sleep deprivation and (2) significant correlation with global mGluR5 availability in both sleep conditions, or with the sleep deprivation-induced increase in mGluR5 availability. These stringent criteria eliminated all but two detected metabolites: myo-inositol and glycine. While the level of myo-inositol was reduced by sleep deprivation (*Figure 4A*), the values in both experimental conditions correlated positively with mGluR5 availability (*Figure 4B and C*). These results reflect a similar intra-subject variation in myo-inositol levels and mGluR5 binding in baseline and after sleep deprivation. In addition, the glycine concentration was enhanced after sleep loss (*Figure 4D*) and the increase was associated with the increased mGluR5 availability (*Figure 4E*). Collectively, these findings may suggest that mGluR5 contribute to sleep regulation by affecting downstream mechanisms of mGluR5-mediated protein phosphorylation and enhanced N-methyl-D-aspartate (NMDA) receptor-mediated signaling.

## Lack of mGluR5 in mice interferes with sleep rebound after sleep deprivation

Given the striking association between mGluR5 availability and markers of sleep homeostasis in humans, sleep-wake regulation was studied in *Grm5* gene (encoding mGluR5) knock-out (KO) mice and heterozygous (HT) and wild-type (WT) littermates. Behavioral states and EEG were quantified over 72 hr under regular light-dark cycles (12:12 hr). A 48 hr baseline period was followed by 6 hr sleep deprivation, starting at light onset (08:00 hr), and 18 hr of recovery.

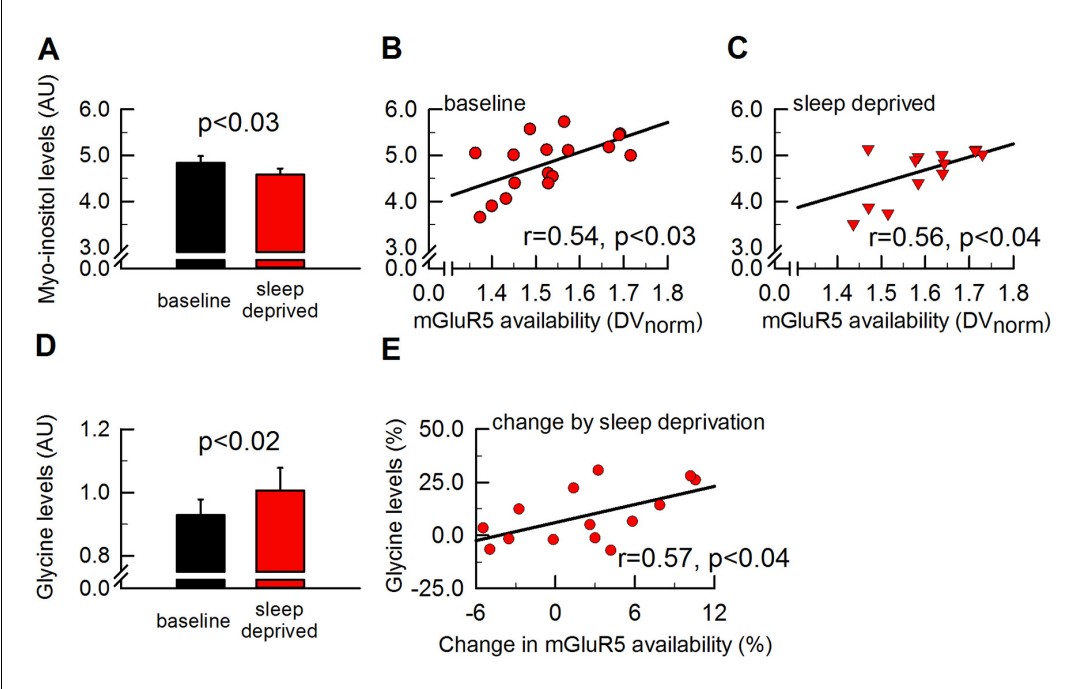

**Figure 4.** Myo-inositiol and glycine levels are altered by sleep deprivation in parallel with mGluR5 availability. To assess cerebral metabolic changes that are altered in synchrony with global mGluR5 availability by sleep deprivation, 17 metabolites measured in the pgACC (see *Figure 2* for further details) were investigated. Whereas neither glutamate, glutamine nor GABA were found to be altered by sleep loss, myo-inositol was decreased by ~5.5% by sleep deprivation (A) and significantly associated with global mGluR5 availability in both sleep conditions (B-C). Glycine was enhanced by ~8.5% by sleep deprivation (D) and the increase in glycine positively correlated with the increase in global mGluR5 availability (E). Error bars represent standard errors of the mean (SEM), p-values indicate paired two-tailed Student's t-tests and Spearman rank correlation coefficients (r).
DOI: https://doi.org/10.7554/eLife.28751.010

Quantification of whole-brain total mRNA by qPCR confirmed the efficiency of the genetic *Grm5* ablation. Thus, the transcript was reduced and completely absent in HT and KO animals (WT: $1.11 \pm 0.02$; HT: $0.61 \pm 0.05$, KO $0 \pm 0$, mean fold change $\pm$ SD; $F_{2,9} = 1338$, p<0.0001, 1-way ANOVA with factor 'genotype'; t-tests Holm-corrected for multiple testing: $p_{all} < 0.0001$; n = 4 per group). Furthermore, qPCR analyses of *Grm5* mRNA expression extracted from cortex, hippocampus and striatum in *Grm5* WT, HT and KO mice were performed after sleep control and sleep deprivation. While *Grm5* expression varied according to allele 'dose', no consistent effect of sleep loss on *Grm5* mRNA expression was observed in neither WT nor HT mice (*Figure 5*). Similarly, other authors recently failed to find differences in mGluR5 protein expression between wakefulness and sleep (*Diering et al., 2017*). Therefore, the sleep deprivation-induced changes in mGluR5 availability found in our human experiment are likely representing a functional synaptic change due to receptor trafficking rather than changes in overall mRNA or protein levels.

In baseline, similar sleep-wake distributions were observed in all *Grm5* genotypes (*Figure 6—figure supplement 1*). In the light phase, the major sleep phase in mice (*Franken et al., 2001*), however, KO animals spent slightly more time in NREM sleep and slightly less time in REM sleep and wakefulness than HT and WT littermates ($F_{2,21} > 7.65$, $p_{all} < 0.004$, factor 'genotype' of two-way ANOVA with factors 'genotype' and 'hour') (*Figure 6—figure supplement 2*). No significant differences among the strains were present in the dark phase ($F_{2,21} < 1.2$, $p_{all} > 0.32$).

A clear sleep-wake phenotype in *Grm5* KO mice became apparent when sleep homeostasis was challenged with sleep deprivation. In the first 6 hr of the recovery period, all genotypes showed an immediate rebound of NREM and REM sleep at the cost of wakefulness (*Figure 6* and *Figure 6—figure supplement 1*). Nevertheless, when the animals entered the dark phase (i.e. 6 to 18 hr after the end of sleep deprivation; ZT 12 to 24), sleep in KO mice was suppressed and wakefulness enhanced despite the preceding sleep deprivation (*Figure 6—figure supplement 1*). After 2 hr into

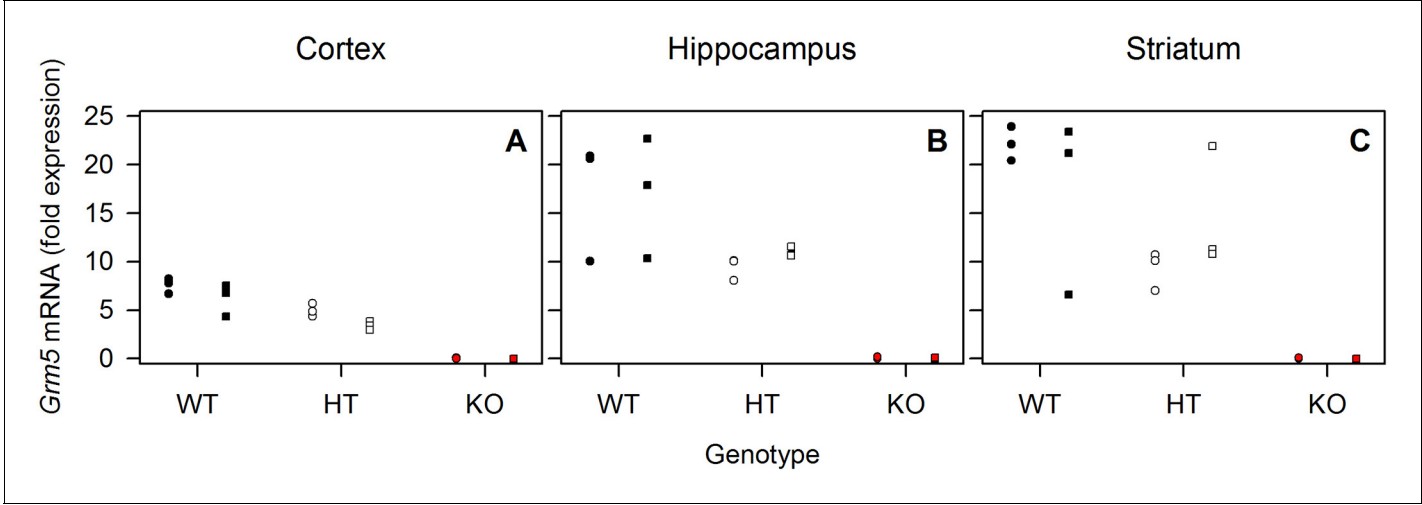

**Figure 5.** Sleep deprivation has no consistent effect on *Grm5* mRNA expression in mice. qPCR of *Grm5* mRNA extracted from cortex (**A**), hippocampus (**B**) and striatum (**C**) of n = 3 WT (black), HT (white) and KO (red) mice in sleep control (circles) and sleep deprivation conditions (squares). Individual data points are depicted. *Grm5* mRNA was expressed in allele dose-dependent manner. Significant '*genotype*' effect ($F_{2,12} > 26.9$, $p_{all} <0.0001$) in two-way ANOVA with '*genotype*' and '*condition*' in all brain regions. No significant main effect of '*condition*' and '*condition*' x '*genotype*' interaction were observed.

DOI: https://doi.org/10.7554/eLife.28751.011

The following source data is available for figure 5:

**Source data 1.** Excel file with one data sheet containing the numerical values of each figure panel (A-C) of *Figure 5*.
DOI: https://doi.org/10.7554/eLife.28751.012

the dark phase, NREM sleep started to decrease, whereas REM sleep remained virtually constant (*Figure 6A and B*). Averaged over the entire recovery period, KO animals lost 29.7 ± 16.7 min of sleep relative to baseline, mainly caused by a reduction in NREM sleep (*Figure 6G*). The KO mice also lacked a REM sleep rebound in the recovery dark phase (*Figure 6B*). In pronounced contrast to the animals without functional mGluR5, HT and WT mice continued to regain NREM and REM sleep in the dark phase (*Figure 6A and B*). By the end of the 18 hr recovery period, these animals had gained 83.5 ± 10.3 min and 82.4 ± 10.5 min of total sleep compared to baseline ($F_{2,21} > 20.68$, $p_{all} <0.0001$, 1-way ANOVA with factor '*genotype*' at ZT 24). In other words, the mean difference between sleep lost in KO animals and sleep gained in WT animals in the recovery period equaled almost two hours (113.2 min).

## Important role for mGluR5 in homeostatic response to sleep deprivation

Next, we asked whether the different sleep-wake distributions across the 72 hr study protocol were accompanied by divergent dynamics of EEG delta (0.75–4.0 Hz) power in NREM sleep (see Materials and methods for details).

In the light phases during baseline, all genotypes showed a similar decline of delta-power in NREM sleep, reflecting the dissipation of homeostatic sleep need (*Figure 7A*). In the dark phases, however, the build-up was greatly attenuated in the KO animals. This attenuation was restricted to the first half of the dark phases (percentiles 1–3), during which the time spent in NREM sleep did not differ among the genotypes (*Figure 6—figure supplement 1*). Thus, the altered dynamics in EEG delta power cannot be explained by differences in time spent in NREM sleep, suggesting a deficient build-up of homeostatic sleep need in wakefulness. This notion was corroborated by substantially reduced delta power in the first two percentiles of the recovery period in KO and HT (to a lesser extent) animals ('*genotype*': $F_{2,18} = 5.723$, $p<0.02$) (*Figure 7B*). No genotype-dependent differences were observed in the recovery dark phase (*Figure 7A*), despite the different distributions of vigilance states (*Figure 6—figure supplement 1*). Although KO mice spent most of the time awake during

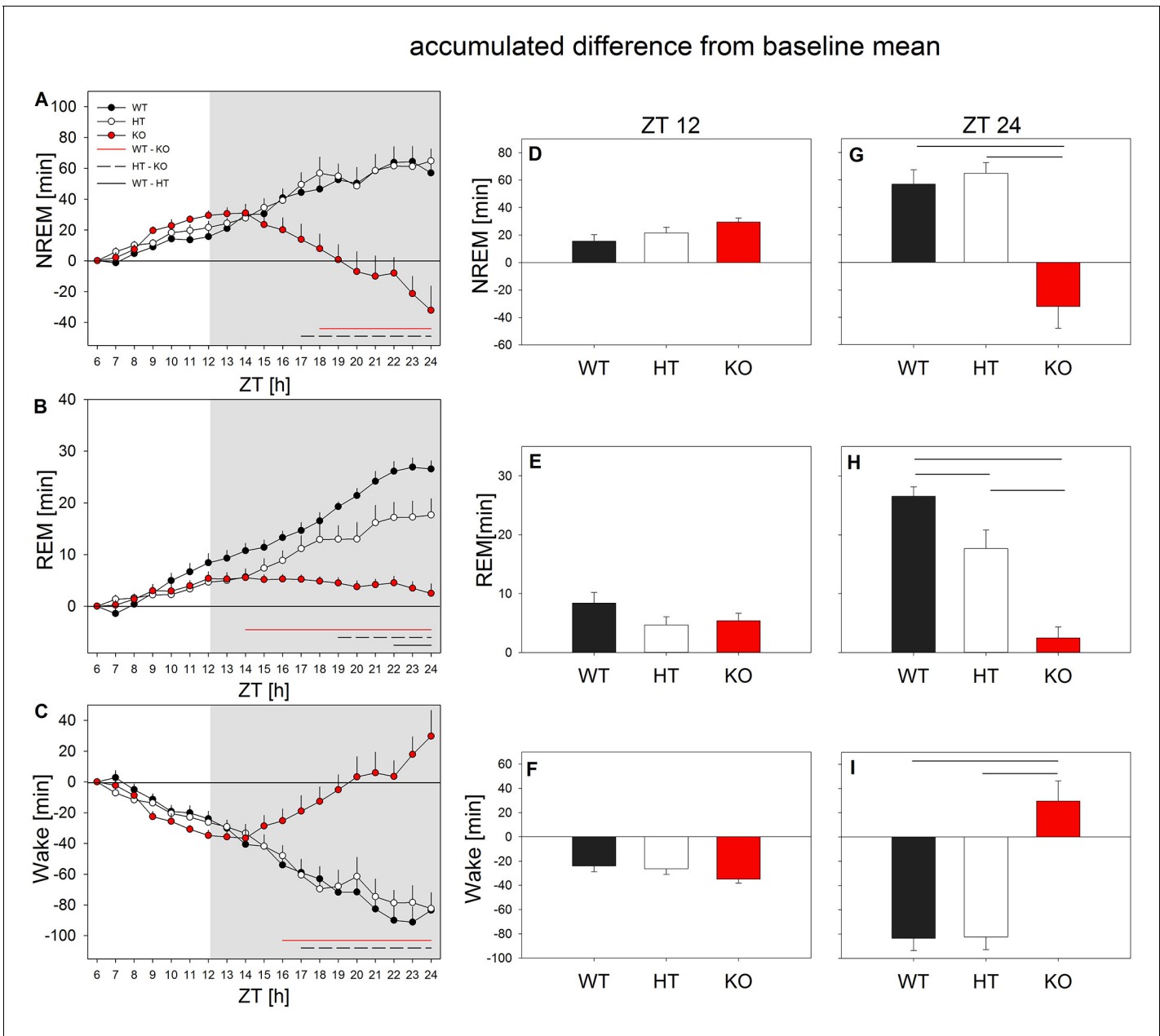

**Figure 6.** mGluR5 modulate the response to sleep deprivation in mice. All mice showed a NREM and REM sleep rebound at the cost of wakefulness in the 6 hr light phase of recovery. During the recovery dark phase, KO mice (red circles) lost NREM and REM sleep, while increasing wakefulness. Left panel: Accumulated difference from baseline (i.e. mean of baselines) across the 18 hr recovery period. Hourly differences in NREM sleep (**A**), REM sleep (**B**) and wakefulness (**C**) from the baseline value were continuously summed up to achieve the accumulated difference from baseline (n = 8 mice/genotype, mean + SEM). Horizontal lines represent Holm-corrected t-tests (p < 0.05), following significant 'genotype' effects in one-way ANOVAs performed at each time point (p < 0.05); WT vs. KO (red line), HT vs. KO (dotted line), WT vs. HT (black line). Middle and right panels: Accumulated mean changes in NREM sleep, REM sleep and wakefulness from baseline to recovery at the end of the 6 hr light phase (ZT12; i.e. accumulated differences of ZT7 – ZT12) (**D-F**) and at the end of the 18 hr recovery period (ZT24; i.e. accumulated differences of ZT7 – ZT24) (**G-I**). No significant effects were observed during the light period. During the dark period of recovery KO mice (red bars) differed significantly from the littermates in their response to sleep deprivation. Horizontal lines represent significant differences between genotypes in Holm-corrected t-test following significant 'genotype' effect in one-way ANOVA: NREM sleep: p < 0.002 for WT vs. KO and p < 0.001 for HT vs. KO; REM sleep: p < 1e-6 for WT vs. KO, p < 0.004 for HT vs. KO and p < 0.04 for WT vs. HT; Wake: p < 0.0004 for WT vs. KO and for HT vs. KO.

DOI: https://doi.org/10.7554/eLife.28751.013

The following source data and figure supplements are available for figure 6:

**Source data 1.** Excel file with one data sheet containing the numerical values of each figure panel (A-I) of *Figure 6*.

*Figure 6 continued*
DOI: https://doi.org/10.7554/eLife.28751.016
**Figure supplement 1.** The effect of mGluR5 genotype across vigilance states and sleep deprivation in mice.
DOI: https://doi.org/10.7554/eLife.28751.014
**Figure supplement 2.** Duration of vigilance states in *Grm5* KO, HT and WT mice at baseline and recovery light and dark phases.
DOI: https://doi.org/10.7554/eLife.28751.015

the initial recovery dark phase (ZT 12 to 16), they lacked a rebound in delta power, demonstrating a severely disturbed sleep homeostatic response to prolonged wakefulness.

## Lack of mGluR5 impairs habituation to environmental challenge after sleep deprivation

To assess the possible behavioral consequences of the sleep phenotype in *Grm5* KO mice, we investigated spatial working memory and exploratory activity in a spontaneous alternation behavior paradigm (for details, see Materials and methods). Different groups of mice were tested in three runs, either after control sleep or after sleep deprivation. Run one was conducted in the active phase at the end of the analyzed recovery period when the differences in accumulated sleep rebound among the genotypes were greatest (ZT 22.5 to 24). Long-lasting consequences of sleep deprivation were assessed at the same circadian time, 1 (run 2) and 6 (run 3) days after run 1.

Working memory, expressed as the percentage of successful entry series (i.e. all three arms of the Y-maze were visited within three consecutive entries), was compromised in KO animals when

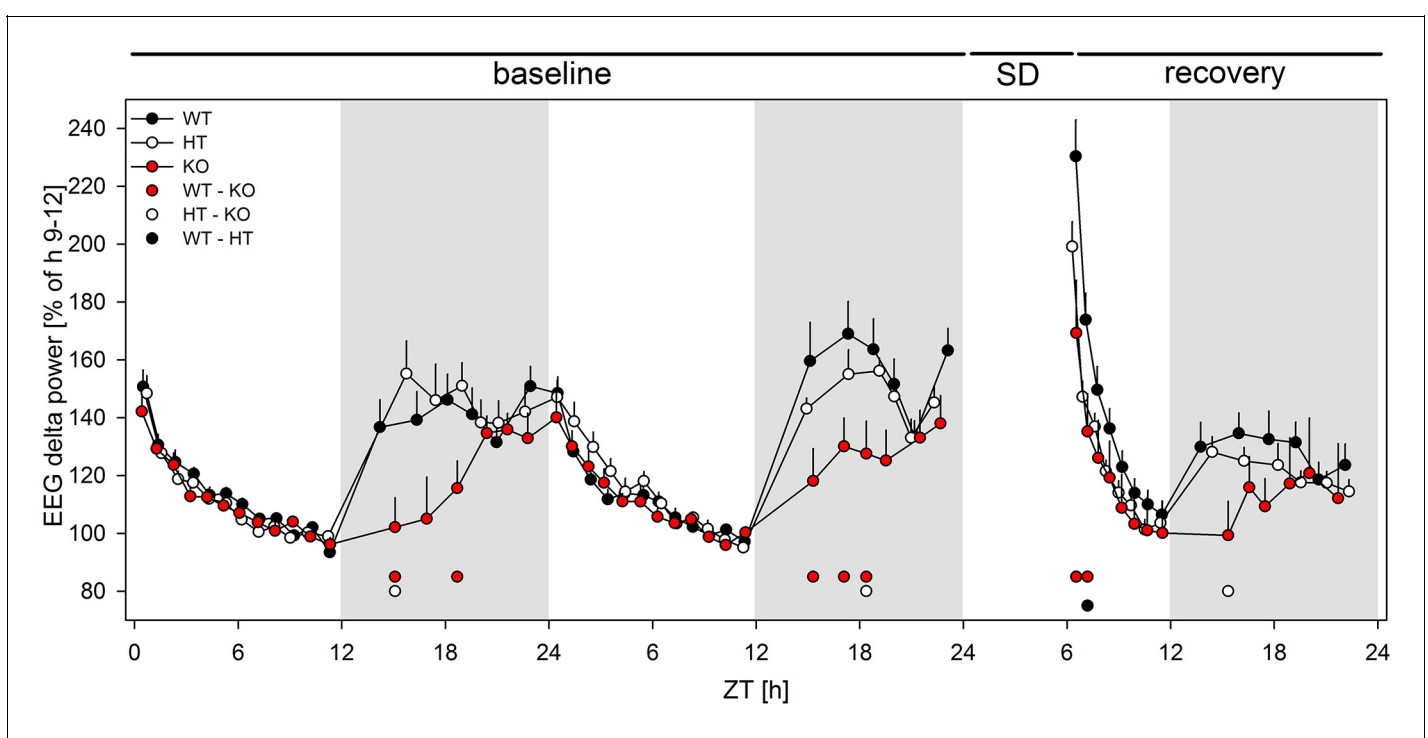

**Figure 7.** Lack of mGluR5 causes marked changes in EEG marker of sleep homeostasis in mice. Time course of the EEG delta power across the 48 hr baseline and 18 hr of recovery (n = 7–8/genotype), expressed as a percentage of the genotype-specific mean reference value at baseline (ZT8 - ZT12). Dots above the x-axes denote significant differences between WT and KO animals (red), HT and KO (white) animals, and WT and HT animals (black), respectively. $p_{all} < 0.05$ (2-sided t-tests), following significant one-way ANOVAs (per time point, factor 'genotype').
DOI: https://doi.org/10.7554/eLife.28751.017

The following source data is available for figure 7:

**Source data 1.** Excel file with one data sheed containing the numerical values of *Figure 7*.
DOI: https://doi.org/10.7554/eLife.28751.018

compared to WT and HT littermates ('genotype': $F_{2,26} > 4.2$, p<0.03) (*Figure 8A–C*). Furthermore, alternation scores were higher than 50% in WT and HT mice ($t_{all}$ >4.6, $p_{all}$ <0.001; one sample t-tests), whereas KO mice performed at chance level (t = 0.95, p=0.18). The scores were similar in control and sleep deprivation conditions ('condition': $F_{1,26} = 0.321$, p=0.58 and 'genotype' x 'condition': $F_{2,26} < 1.8$, p>0.18).

Habituation to the novel environment was assessed by the total number of arm entries (see Materials and methods). Irrespective of experimental condition, KO mice showed reduced exploration at the first encounter with the maze when compared to WT animals (p<0.004) (*Figure 8D*). Arm entries were, thus, normalized to the first exposure, to examine the repercussions of sleep deprivation and genotype on habituation to the maze. In the control condition, all animals markedly reduced exploratory activity from the first to the second and third test runs. This reduction was attenuated when mice were sleep deprived before first maze exposure, especially in KO animals ('condition': run Δ2: $F_{1,40} > 5.0$, p < 0.04; run Δ3: $F_{1,40} > 9.5$, p < 0.004; 'condition' x 'run' interaction: $F_{2,52} > 6.5$,

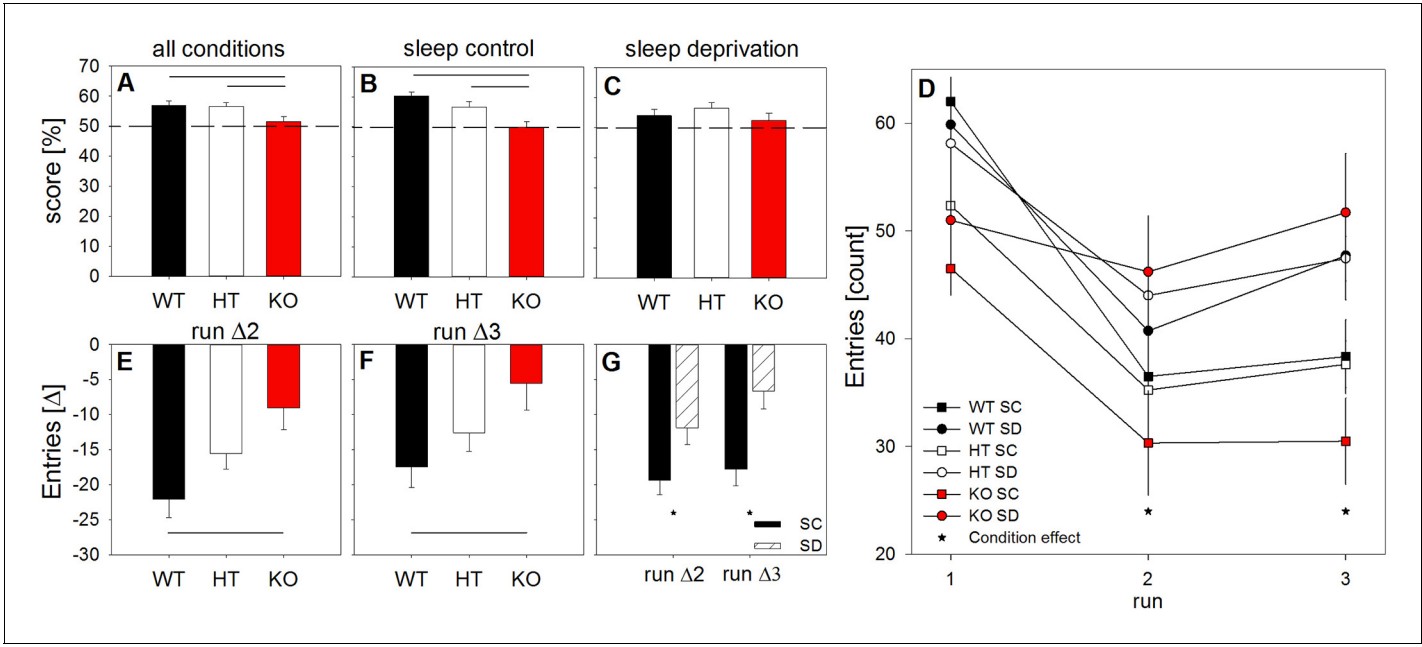

**Figure 8.** Lack of mGluR5 deteriorates working memory and habituation to a novel environment after sleep deprivation. Spontaneous alternation behavior scores in Y-maze expressed as a percentage of successful alternation trials (**A-C**), absolute number of total arm entries (**D**), and differences from the number of entries in the first maze encounter as a measure of habituation to the novel environment (**E-G**). Across all runs and conditions, KO mice (red bars) showed compromised working memory when compared to their littermates ('genotype': $F_{2,26} = 4.29$, p < 0.03). While the WT and HT mice performed above the 50% chance level in all conditions, the KO animals did not (**A-C**). Horizontal black lines indicate significant differences between genotypes (p < 0.03; Holm-corrected t-tests). In run 1, KO animals showed fewer total arm entries than their WT littermates. When studied after sleep deprivation (circles), mice displayed higher overall exploration than mice under control conditions (squares) ('condition': $F_{1,26} > 12.1$, p < 0.002). This effect of sleep deprivation on arm entries was modulated by genotype and test run ('run', 'genotype' x 'run', and 'run' x 'condition': $F_{all} > 3.0$, $p_{all} < 0.03$). In runs 2 and 3, sleep-deprived mice lacking mGluR5 produced significantly more arm entries than control mice without mGluR5 (stars: $t_{all} < -2.3$, $p_{all} < 0.05$; Welch two-sample t-tests) (**D**). Normalization of the number of arm entries in runs 2 and 3 to run 1, confirmed the effects of sleep deprivation ('condition': $F_{1,40} > 5.0$, $p_{all} < 0.04$) and genotype ($F_{2,40} > 4.0$, $p_{all} < 0.03$) on habituation across the three test trials. More specifically, sleep deprivation attenuated habituation to the Y-maze compared to the sleep control condition, as reflected in the reduced decrease in arm entries in runs 2 and 3 relative to run 1 (stars: $p_{all} < 0.03$; two-samples t-tests) (**G**). Finally, the sleep-deprived KO animals showed a less pronounced reduction in exploratory activity than the sleep-deprived WT animals. Horizontal black lines indicate significant differences between WT and KO mice ($p_{all} < 0.05$) (**E-F**). All data points represent means ± SEM (n = 6–10 mice per condition and genotype).

DOI: https://doi.org/10.7554/eLife.28751.019

The following source data and figure supplement are available for figure 8:

**Source data 1.** Excel file with one data sheet containing the numerical values of each figure panel (A-G) of *Figure 8*.

DOI: https://doi.org/10.7554/eLife.28751.021

**Figure supplement 1.** Change of exploratory activity across trials.

DOI: https://doi.org/10.7554/eLife.28751.020

p < 0.003; 'genotype': $F_{2, 40} > 4.0$, $p_{all} < 0.03$; 'genotype' x 'run' interaction: $F_{4,52} > 3.0$, $p < 0.03$) (**Figure 8E–G**). After prior challenge with prolonged wakefulness, these animals showed stable exploratory behavior across all test runs, suggesting that *Grm5* KO mice are more impaired by sleep deprivation than WT and HT littermates and that the impairment persists for up to 1 week (**Figure 8—figure supplement 1**).

## Discussion

This study provides compelling novel evidence for the notion that mGluR5 and the mGluR5 signaling cascade are an important part of the molecular mechanisms underlying the regulation of sleep need in humans and mice. In humans, increased mGluR5 availability after sleep deprivation was tightly associated with increased propensity to fall asleep during brain imaging, as well as delta and low frequency (<1 Hz) EEG activity in NREM sleep, reliable neurophysiological hallmarks of sleep homeostasis. Magnetic resonance spectroscopy further identified associated changes in brain myo-inositol and glycine levels, providing evidence for sleep loss-induced modifications in the downstream signaling cascade of mGluR5. Finally, the build-up of delta power during wakefulness in baseline and recovery periods in mice devoid of functional mGluR5 was severely disturbed, particularly during the dark phases of baseline and recovery periods. Together, the findings highlight in convergent translational fashion that mGluR5 likely contribute to functional aspects of sleep. For example, enhanced <1 Hz EEG activity in early NREM sleep improves memory functions in humans (**Marshall et al., 2006**), whereas in mice the mGluR5 signaling complex is important for the consolidation of contextual memory in vitro and in vivo (**Huber et al., 2001**; **Lu et al., 1997**).

### Functional mGluR5 availability predicts EEG low (<1 Hz) activity in NREM sleep

The slow (<1 Hz) rhythm in the human EEG and its cellular counterpart, the slow oscillation, are characterized by brief sequences of membrane depolarization and intense neuronal firing (up-states), followed by membrane hyperpolarization and neuronal silence (down-states) (**Achermann and Borbély, 1997**; **Steriade et al., 1993**; **Steriade, 1997**; **Vyazovskiy et al., 2009**). In vitro and in vivo data suggest that mGluR1 and mGluR5 contribute to the regulation of the slow oscillation and the associated fluctuations in the membrane potential between up-states and down-states (**Hays et al., 2011**; **Hughes et al., 2002**). While other mechanisms regulating cortico-thalamic interactions and excitability of cortical neurons are likely also involved, these observations are consistent with the striking positive association found in this study between global mGluR5 availability and <1 Hz EEG activity, suggesting that mGluR5 may be important for maintaining or generating this slow rhythm. The slow oscillation and the corresponding down-state may provide a period of reduced synaptic inputs and be important for neuronal maintenance and rest (**Vyazovskiy and Harris, 2013**). Our data indicate that these rest periods are regulated or gated by mGluR5. Intriguingly, the slow <1 Hz rhythm has been associated with increased blood flow in precuneus, posterior cingulate, as well as medial frontal, parietal, and central gyri (**Dang-Vu et al., 2008**). These brain regions are almost identical with those where we observed the strongest associations between mGluR5 availability and EEG <1 Hz activity. In fact, mGluR5 availability in parietal cortex and precuneus explained as much as 64% and 70% of the intra-individual variance in <1 Hz EEG activity in baseline and recovery sleep.

### Functional mGluR5 availability predicts rebound in low-frequency EEG oscillations in NREM sleep after sleep deprivation

Slow-wave or delta activity in NREM sleep is the best established marker of homeostatic sleep need (**Achermann and Borbély, 2011**). Here, we found a close association between global mGluR5 availability and initial EEG delta activity in NREM sleep in baseline and after sleep deprivation. Furthermore, the sleep deprivation-induced increase in global mGluR5 availability was positively correlated with increased subjective sleepiness (**Hefti et al., 2013**), reduced capacity to stay awake during brain imaging, as well as the rebound in delta activity in NREM sleep after prolonged wakefulness. These findings strongly indicate that mGluR5 activation constitutes a molecular mechanism keeping track of sleep-wake history. This notion is further supported by the altered sleep-wake distribution and dysfunctional dynamics of EEG delta power in NREM sleep in *Grm5* KO mice.

## Sleep deprivation likely affects downstream signaling pathway of mGluR5 activation

Homer1a, the best established molecular substrate of homeostatic sleep-wake regulation today (*Mackiewicz et al., 2008*; *Maret et al., 2007*), selectively uncouples mGluR5 from the intracellular effector mechanisms such as the IP$_3$ pathway (*Diering et al., 2017*; *Ménard and Quirion, 2012*; *Tu et al., 1998*). In this way, Homer1a buffers in activity-dependent manner the mGluR5-dependent release of calcium from intracellular stores (*Bottai et al., 2002*). The two metabolites found to be associated with mGluR5 availability and sleep deprivation, myo-inositol and glycine, are both linked to the mGluR5-Homer1a-IP$_3$ signaling cascade (*Berridge, 1984*; *Moghaddam and Javitt, 2012*). Myo-inositol is the most abundant inositol derivative in the brain and a structural precursor of IP$_3$ (*Croze and Soulage, 2013*). The positive association with mGluR5 availability suggests that the two molecules are tightly linked in the human brain. Moreover, myo-inositol was previously shown to be inversely coupled to neuronal activity (*Xu et al., 2005*). A plausible interpretation of the present data could be that the sleep deprivation-induced reduction in myo-inositol is caused by increased activation of mGluR5 after sleep loss, which triggers the formation of IP$_3$ at the cost of myo-inositol. Future studies are warranted to corroborate this possible underlying mechanism.

Glycine is an important allosteric modulator of glutamatergic NMDA receptors (*Johnson and Ascher, 1987*) and activation of mGluR5 triggers enhanced activity of these receptors (*Awad et al., 2000*; *Conn and Pin, 1997*). The concomitant sleep loss-induced rise in glycine and the availability of mGluR5 suggest that NMDA receptor activation is enhanced after sleep deprivation. Indeed, in vitro electrophysiological analyses and immunoblotting of purified synaptosomes demonstrated that insufficient sleep alters NMDA receptor subunit composition and functions, whereas recovery sleep reverses these sleep deprivation-induced changes (*Kopp et al., 2006*). Furthermore, studies in rats revealed that glycine in low doses promotes wakefulness and reduces sleepiness (*Bannai et al., 2012*), whereas high doses of glycine rather promote NREM and REM sleep in a NMDA receptor-dependent manner (*Kawai et al., 2015*). In view of the diverse effects of glycine on wakefulness and sleep, the elusive mechanisms underlying its rise by sleep deprivation, as well as glycine's varying functions in different neuronal circuits (*Giber et al., 2015*; *Zeilhofer, 2005*) the present data need to be interpreted with caution. Nevertheless, the spectroscopic findings may indicate that the mGluR5-associated Homer1a-IP$_3$ and glycine-NMDA receptor pathways importantly contribute to the molecular machinery that regulates sleep-wake homeostasis in humans and open up promising new avenues for future research.

It may be noteworthy that the endogenous ligand of mGluR5, glutamate, and the related metabolites, glutamine and γ-amino-butyric acid (GABA), were not altered by sleep deprivation nor associated with the availability of mGluR5 ($t_{15} < 0.67$, $p_{all} > 0.5$; data not shown). This observation suggests that the association between functional mGluR5 availability and sleep deprivation likely does not reflect upstream changes in glutamatergic signaling. Nevertheless, because PET and $^1$H-MRS imaging were slightly displaced and the spectroscopic data were acquired from a single voxel, future studies are needed to establish the generalizability of this notion.

## Genetic loss-of-function of mGluR5 in mice causes dysregulation of sleep-wake patterns and sleep EEG delta activity

The baseline sleep-wake pattern in mice without functional mGluR5 was similar as in their littermates, although a slight variation in the distribution of NREM sleep, REM sleep and wakefulness during the light phase was present. When sleep homeostasis was challenged by sleep deprivation, however, a clear dysregulation of wakefulness and sleep became apparent. The KO animals not only lacked the normal rebound in NREM and REM sleep in the recovery dark phase, they even lost roughly half an hour of NREM sleep when compared to baseline. This is in striking contrast to more than 1 hr of sleep gained in HT and WT mice by the end of the recovery period. To our knowledge, no comparable sleep-wake phenotype has ever been described in the literature in an animal model before. The data corroborate that functional mGluR5 are necessary for maintaining the normal physiologic homeostatic sleep response to sleep deprivation.

Already in baseline, the KO mice showed a markedly attenuated build-up in delta power during spontaneous wakefulness, although the distribution of vigilance states was comparable to the other genotypes. When challenged with sleep deprivation, both KO and HT animals showed an reduced

build-up of delta power, resulting in lower values at the beginning of the recovery light phase when compared to WT littermates. In accordance with our findings, a diminished rebound in NREM sleep EEG delta power after prolonged wakefulness in *Grm5* KO animals was reported previously (*Ahnaou et al., 2015b*). Nevertheless, the profound changes in sleep amounts and the dynamics of EEG delta power observed here were not found in the previous study. The discrepancy between the studies may be due to several methodological differences, in particular the use of an automated sleep deprivation technique by the other authors which induces forced locomotion. Forced locomotion causes stereotypic behavior unequal to normal wakefulness and the dissipation of sleep pressure (*Fisher et al., 2016*). Additionally, unlike the previous report, we also studied a HT group and found an intermediate phenotype between the WT and KO mice. This gene dose-effect relationship further highlights the importance of mGluR5 for the sleep-wake-dependent regulation of delta activity in NREM sleep.

## Lack of mGluR5 compromises adjustment to novel environment after sleep deprivation

Consistent with previously reported deficits in various learning and memory paradigms (*Jia et al., 1998*; *Manahan-Vaughan and Braunewell, 2005*), *Grm5* KO mice performed worse than WT and HT littermates in a Y-maze working-memory task. In addition, KO animals initially explored less and were compromised in adapting exploratory behavior to the novel environment when sleep deprived. In agreement with the literature (*Hagewoud et al., 2010*; *Niijima-Yaoita et al., 2016*), the sleep deprivation challenge had no consistent effect on initial arm entries. One and 6 days after initial testing, however, the sleep deprived animals reduced exploratory activity to a lesser extent than the non-sleep deprived controls. Thus, acute sleep deprivation appears to induce a long-lasting impairment of the normal habituation to a novel environment (*Bolivar, 2009*). Mice lacking functional mGluR5 were particularly sensitive to this impairment, suggesting that mGluR5 contribute to the beneficial role of sleep to habituate to stressful conditions. Whether this phenotype is a consequence of compromised mGluR-dependent LTD (*Lüscher and Huber, 2010*; *Manahan-Vaughan and Braunewell, 1999*) remains to be determined.

In conclusion, our study provides converging translational evidence that increased mGluR5 availability following sleep loss in humans is associated with objective markers of sleep need and that lack of functional mGluR5 in mice severely affects sleep homeostasis. The question remains whether increased mGluR5 availability is a compensatory mechanism to promote wakefulness in the sleep-deprived state, signals the necessity to sleep, or both. Recent studies investigating the effects of positive and negative allosteric modulators of mGluR5 in rats may help to tackle this question (*Ahnaou et al., 2015a*). It was found that the positive allosteric modulator, ADX47273, promoted wakefulness and reduced NREM sleep and total sleep time. On the contrary, the mGluR5 negative allosteric modulator, MPEP (2-methyl-6-(2-phenylethynyl)pyridine;hydrochloride), increased total sleep time and sleep efficiency. Thus, sleep-wake-dependent changes in mGluR5 signaling may aid or facilitate sustained wakefulness and the proper homeostatic build-up of sleep propensity during wakefulness as reflected in EEG delta power in NREM sleep. At the same time, mGluR5-dependent mechanisms may promote and maintain deep sleep rich of slow waves. As such, mGluR5 could provide a promising new target for sleep-wake enhancing compounds, which may be beneficial in treating sleep-wake disorders such as hypersomnia or insomnia. Furthermore, behavioral studies will have to determine whether interventions targeting mGluR5 may promote sleep-associated brain functions such a memory consolidation and stress resilience.

## Materials and methods

All experimental procedures were conducted in accordance with the declaration of Helsinki (1964) and approved by the cantonal (ethics committee for research on human subjects of the canon of Zurich [Reference Nr. EK-Nr. 786] and ethics committee of the State of Vaud Veterinary Office [No. 2699.0]) and Swiss federal authorities for research on human (Swiss Federal Institute of Public Health, Reference Nr. 464-0002-6/08.005701) and animal subjects.

## Studies in humans

### Study participants and pre-experimental procedure

A total of 26 healthy young men completed a 2-week study after providing written informed consent, and consent to publish. All study participants fulfilled strict inclusion criteria with respect to sleep quality and psychological wellbeing, and abstained from medication and drug use (*Hefti et al., 2013*). Three days before each experimental block, participants consumed neither caffeine nor alcohol and stringently adhered to an 8 hr sleep/16 hr wake schedule, verified by measuring caffeine in saliva, breath-alcohol levels, wrist-actigraphy, and sleep logs.

### Sleep deprivation and imaging protocol

The experimental protocol was previously explained in detail elsewhere (*Hefti et al., 2013*). In short, all subjects completed in randomized, cross-over fashion two experimental blocks consisting of baseline and sleep deprivation conditions. To ensure sustained wakefulness, subjects were constantly supervised throughout the protocol. Eight-hour sleep episodes in baseline and recovery nights (23:00 – 07:00 [n = 9] or 00:00 – 08:00 [n = 17]) before and after prolonged wakefulness were recorded with polysmonography.

### Positron emission tomography (PET) image acquisition

After ~9 (16:39 PM ±8.23 min) and ~33 (16:30 PM ±7.40 min) hours of (prolonged) wakefulness, a 60 min positron emission tomography (PET) scan with the highly selective mGluR5 radioligand, [11]C-ABP688, was performed at the Division of Nuclear Medicine, University Hospital Zürich (see *Figure 1A* for study protocol). Because of technical problems, three subjects had to be excluded from the PET analyses. Tracer synthesis and three-dimensional PET brain imaging with 2.3 × 2.3 × 3.2 mm voxel resolution was carried out using a previously validated bolus/infusion protocol (*Ametamey et al., 2006*; *Ametamey et al., 2007*; *Burger et al., 2010*; *Deschwanden et al., 2011*) on two GE Healthcare PET scanners (DVCT PET/CT or DSTx PET/CT scanner) (*Hefti et al., 2013*). All measurements in each study participant were always conducted on the same scanner. Image processing consisted of within-subject rigid-matching motion correction, as well as spatial normalization of averaged frames 17 to 19 (45–55 min) to the Montreal Neurological Institute (MNI) template brain in PMOD software package, version 3.2 (PMOD Technologies, Zürich, Switzerland). Quantification of the PET images was achieved by dividing regional radioactivity concentration values in nine predefined volumes of interest (VOIs) with high mGluR5 expression and suggested involvement in sleep-wake regulation with the corresponding value in the cerebellum ($C_{t[Cb]}$) to obtain $DV_{norm}$ ($DV_{norm} = C_{t[VOI]}/C_{t[Cb]}$). The VOIs included left and right medial superior frontal cortex, orbito-frontal cortex, dorso-lateral prefrontal cortex, anterior cingulate cortex, inferior parietal cortex, precuneus, insula, striatum, and parahippocampal gyrus. In addition, global whole-brain changes in mGluR5 availability were investigated.

Injected dose of radioactivity (582 ± 18.2 [baseline] vs. 568 ± 18.5 MBq/ml [sleep deprivation]), specific activity at the end of the synthesis (127 ± 14.5 vs. 107 ± 11.4 GBq/μmol), mass of cold compound (6.39 ± 0.65 vs. 6.92 ± 0.59 nmol) and cerebellar standard uptake values (826 ± 29.7 vs. 788 ± 24.9 g/ml) did not differ between the conditions ($p_{all} > 0.1$, 2-tailed paired $t$-tests).

### Acquisition of magnetic resonance spectroscopy (MRS) data

Immediately following PET imaging, a subsample of 17 study participants underwent a magnetic resonance (MR) imaging session in a Philips Achieva 3T whole-body MR unit, equipped with transmit/receive head coil (Philips Healthcare, Best, Netherlands). During MR imaging, subjects were instructed not to fall asleep. To ensure maintained wakefulness, subjects were instructed to press a button on a response box with a frequency of roughly 1 Hz. If subjects ceased pressing, they were alerted immediately via intercom until response-pressing continued. Because only a single MR-scanner was available, two subjects were always recorded consecutively,~45 min or ~1 hr 40 min after completion of the PET scan. Anatomical whole-brain T1-weighted three-dimensional fast gradient echo MRI scans (180 slices, FOV = 220 mm, matrix = 224 × 224 reconstructed to 256 × 256, voxel size = 0.98 × 0.98 × 1.5 mm) were obtained to exclude subjects with morphological abnormalities, in order to co-register MR and PET images for accurate cerebellum delineation and to accurately position the [1]H-MRS voxels.

Proton magnetic resonance spectroscopy ([1]H-MRS) data were acquired from a single 9 ml voxel (25 × 18 × 20 mm) (*Figure 1*) in the bilateral pregenual anterior cingulate cortex (pgACC) using a birdcage transmit-receive head coil with a maximum radiofrequency field strength ($B_1$) of 20 μT (*Ernst et al., 2017*). The [1]H-MRS was performed using a maximum echo-sampled 2-D J-resolved point-resolved spectroscopy (JPRESS) sequence (relaxation time ($T_R$) of 1600 ms, echo time ($T_E$) ranging from 26 to 224 ms with step size of 2 ms, 100 encoding steps, eight averages per step) with variable power and optimized relaxation delays (VAPOR) water and interleaved inner volume suppression. [1]H-MRS acquisitions lasted 22 min and allowed for the quantification of 17 metabolites. Besides the three metabolites directly associated with glutamatergic neurotransmission: glutamate, glutamine and γ-aminobutyric acid (GABA); an additional 14 metabolites were quantified, including: total creatine and phosphocreatine, n-acetyl aspartate (NAA), N-acetylaspartylglutamic acid (NAAG), total choline, myo-inositol, scyllo-inositol, glucose, lactate, taurine, glycine, glutathione, phosphoethanolamine, aspartate, and ascorbic acid. MR-spectra quantification was improved by ProFit 2.0 fitting, a method which has been extensively validated (*Fuchs et al., 2014*). Prototypical projections of 2-D J-PRESS spectra including spectral fit and residuals of the pgACC in sleep control and sleep deprived condition for one representative individual are depicted in *Figure 1B and C*.

Because the two [1]H-MRS scans were always performed consecutively, the effect of scan-order and sleep deprivation (condition) was assessed using two-way repeated measure ANOVAs. Applying internal water as a reference, total creatine was observed to be significantly altered by scan-order ($F_{1,15}$ = 4.91, $p < 0.05$). Because of this instability in creatine, metabolite levels were all referenced to internal water. The water referenced values were corrected for segmentation-based volume tissue composition and relaxation (*Gasparovic et al., 2006*). Individual T2 relaxation correction were performed in ProFit 2.0, with the resulting metabolite concentrations reported in arbitrary units (*Fuchs et al., 2014*; *Hulka et al., 2016*).

To evaluate the quality of the ProFit 2.0 fit, the Cramér-Rao lower bounds (CRLBs) was used as an internal control for each metabolite peak. Metabolite estimates with CRLBs > 20% were excluded. Because of movement artifacts or CRLBs > 20%, one spectrum had to be excluded in the sleep deprived condition. The quality criterion based on the CRLBs resulted in exclusion of a few data points. This applied especially to the analysis of the smaller GABA peak, in which n = 13 (control condition) and n = 10 (sleep deprivation condition) could be included. For the analyses of glutamate, glutamine, myo-inositol and glycine, n = 17 (control condition) and n = 14 (sleep deprivation condition) were included.

## EEG and polysomnographic recordings

Continuous polygraphic recordings were conducted during PET scans and in all experimental nights. The EEG, electrooculogram (EOG), submental electromyogram (EMG), and electrocardiogram (ECG) were recorded with Rembrandt Datalab, Version 8 (Embla Systems, Planegg, Germany) using an Artisan polygraphic amplifier (Micromed, Mogliano Veneto, Italy). During PET image acquisition, subjects were instructed not to fall asleep and in case of sleep-like EEG activity, they were alerted via intercom. The EEG recordings were started shortly before initiation of PET imaging. The recording durations were virtually identical in the two conditions: baseline: 67.3 ± 1.0 min; sleep deprivation: 67.3 ± 0.7 min (p > 0.95). The amounts of intermittent N1 sleep expressed as a percentage of measurement time were analyzed.

The analog EEG signals were sampled at 256 Hz and conditioned by high-pass (−3 dB at 0.15–0.16 Hz) and low-pass filtering (−3 dB at 67.2 Hz). The EEG was recorded from one referential (C3M2) and eight bipolar derivations; the data of the C3M2 derivation are reported here. Sleep stages were visually scored in 20 s epochs according to standard criteria (*Iber et al., 2007*). Four-second EEG spectra (fast Fourier transform [FFT] routine, Hanning window, 0.25 Hz resolution; 0–20 Hz) were calculated with MATLAB (MathWorks Inc., Natick, MA), averaged over five consecutive epochs, and matched with scored sleep stages. Arousal- and movement-related artifacts were visually identified and eliminated. Mean slow-wave activity (SWA; EEG power within 0.5–4.5 Hz) and power in the frequency range of the sleep slow oscillation (power within 0.25–1.0 Hz) in the first NREM sleep (stages N1–N) episodes (*Feinberg and Floyd, 1979*) were calculated in each participant in the baseline night of the first study block and in the recovery night after sleep deprivation.

Unless otherwise specified, statistics from EEG data are performed on logarithmic base 10 transformed values.

## Studies in mice

The *Grm5*[-/-](KO) mice were generated as previously described (*Jia et al., 1998*) and backcrossed to C57BL/6J mice. Experiments were carried out with adult (10–12 weeks) male *Grm5*[+/+] (WT), *Grm5*[+/-] (HT) and KO littermates obtained by heterozygous breeding. Mice were group- or single housed according to the experimental procedure in polycarbonate cages (31 × 18 × 18 cm) in a temperature- (25°C) and humidity-controlled (50-60%) environment under a 12:12 hr light dark cycle (lights on 8:00, 70-90 lux). Food and water were available *ad libitum*.

### Sleep deprivation procedure

The EEG/EMG signals were recorded for 48 hr of undisturbed baseline (i.e. baseline 1 and baseline 2, 24 hr each). A 6 hr sleep deprivation started at light onset (ZT [zeitgeber] 0 to 6) of the third day and was achieved by so called 'gentle handling'. During this procedure, animals are not handled but the bedding material is gently moved, the cage tapped, or novel objects introduced as soon as behavioral signs of sleep appeared. In addition, mice are provided with a clean cage halfway during the sleep deprivation (ZT 3), which provides additional stimulation. The sleep deprivation was followed by a 18 hr recording period during which recovery from sleep deprivation was quantified (n = 8 mice/genotype).

### EEG/EMG implantation and recordings

Surgery for fronto-parietal EEG recordings was performed under deep ketamine/xylazine anesthesia (intraperitoneal injection, 75 and 10 mg/kg at a volume of 10 ml/kg) at 10-12 weeks of age. EEG/EMG signals were obtained as previously described (*Mang and Franken, 2012*). Briefly, six gold-plated mini-screws (1.1 mm diameter) were implanted over frontal and parietal cortices. Over the right hemisphere, two screws soldered to the recording leads prior to implantation, served as frontal (1.5 mm anterior to bregma, 1.7 mm lateral to midline) and parietal (1.0 mm anterior to lambda, 1.7 mm lateral to midline) electrodes. The remaining four screws served as anchors. For EMG recordings two gold wires were inserted into the neck muscle. Anchor screws, EEG and EMG electrodes were fixed to the skull by using dental cement. The incision was closed by stitching and animals received a 10 mg/kg dose of analgesic (Flunixine). Animals were allowed 5 days to recover, followed by a 6-day habituation period, to adapt to the connecting cable. EEG/EMG signals were recorded with Somnologica (Somnologica Science 3.3.1.1529, Medcare), amplified, analog-to-digital converted (2 kHz) and down-sampled to 200 Hz. Power spectral analysis between 0–100 Hz was performed with FFT of 4 s epochs of frontal-parietal differential EEG recordings, yielding a frequency resolution of 0.25 Hz. The vigilance states wakefulness (W), REM sleep and NREM sleep were determined for consecutive 4 s epochs using standard criteria (*Mang and Franken, 2012*). Epochs containing EEG artifacts were marked and excluded from spectral analysis.

### Analyses of vigilance states

The minutes spent in each vigilance stage per hour across the 72 hr experiment were analyzed with two-way repeated measures ANOVAs (n = 8 mice/genotype; factors 'genotype' and 'hour'), to detect genotype-dependent differences in the light and dark phases in baseline (12 hr each) and recovery (6 hr light and 12 hr dark). One-way ANOVAs with the factor 'genotype' were carried out on hourly values as post-hoc analysis after the overall ANOVA reached significance (p<0.05 for the factor 'genotype' or interaction 'genotype' x 'hour').

To investigate differences among genotypes in their response to sleep deprivation, statistical analyses in light and dark periods of recovery were performed on the mean baseline relative values, calculated in each individual mouse with two-way repeated measure ANOVA (factors 'hour' and 'genotype').

To describe the progression of rebound behavior after sleep deprivation, accumulated change in vigilance states (recovery – baseline) was calculated and depicted. One-way ANOVAs with the factor 'genotype' were performed on hourly values. All significant genotype effects were further decomposed by performing t-tests, Holm-corrected for multiple testing, as post-hoc analysis.

## Spectral analysis

For spectral analysis, artifacts and state transition epochs were excluded, that is, only epochs preceded and followed by epochs of the identical vigilance state were considered. Data were normalized and weighted according to contributing vigilance states as previously described (*Franken et al., 1999*; *Mang et al., 2016*). In brief, absolute spectral power density (PSD) between 0.75 and 49 Hz in 0.25 Hz bins was measured for each vigilance state and condition. To account for inter-individual differences in total EEG power, overall reference values were calculated (*Franken et al., 1999*; *Mang et al., 2016*). This was done by calculating the area under the PSD curve per vigilance state for baseline and multiplying it by the percentage of contributing epochs per 24 hr. The sum of weighted values corresponds to the total power reference values used for further analysis. Absolute state specific power (i.e. Wake, NREM, REM separately), was divided by the total power reference value of the respective mouse, to obtain the relative power data used for statistical analysis.

## Time course analysis of EEG delta power

The time course of delta power (0.75–4 Hz) in NREM sleep was computed as previously described (*Franken et al., 1999*; *Maret et al., 2007*). To investigate the temporal progression of delta power, values were computed relative to the last 4 hr of light phase (i.e. ZT 8 to 12 defined as 100%) in baseline (mean of baseline days 1 and 2 per mouse). This interval corresponds to the period with lowest delta power (*Franken et al., 2001*; *Mang and Franken, 2012*).

The number of NREM sleep epochs in the light and dark phases were quantified for all recording days. To adjust for the different occurrence of NREM sleep across the light and dark cycle, recording segments were subdivided into intervals (12 intervals for 12 hr light phases, 6 intervals for 12 hr dark phases, and eight intervals for the 6 hr recovery light phase subsequent to sleep deprivation; see also *Figure 7*) with an equal number of contributing NREM epochs for all intervals of the respective segment. Normalized mean delta power was calculated by dividing mean absolute delta power with the respective reference value for each interval and mouse. The ZT-time of each interval was calculated as the mean ZT-time of the contributing NREM epochs. Statistical analysis was done by 1-way ANOVAs, with factor *'genotype'* for each time point (i.e. interval), followed Holm-corrected t-test (n = 7–8 mice/genotype; due to aberrant spectral power in one mouse in the recovery period [>2 standard deviations from the mean], this animal was excluded from the quantitative EEG analyses).

## Quantitative real-time PCR

Whole brain total RNA of undisturbed WT, HT and KO mice (n = 4) and total RNA of cortex, striatum and hippocampus of sleep deprived mice and their littermate controls (n = 4) was extracted using RNase Lipid Tissue Midi Kit (Qiagen) and treated with RNase-free DNase (Qiagen, Hilden, Germany). RNA content and quality were controlled by NanoDrop for quality control ration 260/280 ~2. Because some extracts from dissected brain areas did not fulfill these criteria they had to be excluded. n = 3 per group could be used for analyses.

From 1 µg of sample RNA and controls (no-template control and no-enzyme control) cDNA was produced by reverse transcription PCR using random primers. cDNA was subsequently used for quantitative PCR (TaqMan) probing for *Grm5* (probe/primer assay Applied Biosystems) and the reference genes EEF1a1 (eucaryotic elongation factor 1a1), TBP (TATA-box-binding protein) and Rps9 (ribosomal protein S9). Probes and primers for reference genes were designed in-house and produced by Microsynth (Balgach, Switzerland). Quantification of the amplification was performed in triplicates, that is, technical replicates, in QuantStudio 6Flex Real-time PCR System (Thermo Fisher Scientific, Massachusetts). Fold-expression relative to the reference genes was analyzed by one-way ANOVA with factor *'genotype'* for whole brain RNA which has been extracted from mice under undisturbed conditions, and by two-way ANOVA with factors *'genotype'* x *'condition'* for dissected brain areas in sleep deprived and sleep control mice.

Probe and primer sequences for quantitative real-time PCR: EEF1a1 (forward primer: 5'-CCTGGCAAGCCCATGTGT-3'; reverse primer: 5'-TCATGTCACGAACAGCAAAGC-3'; probe: 5'-TGAGAGCTTCTCTGACTACCCTCCACTTGGT-3'), TBP (forward primer: 5'-TTGACCTAAAGACCATTGCACTTC-3'; reverse primer: 5'-TTCTCATGATGACTGCAGCAAA-3'; probe: 5'-TGCAAGAAATGCTGAATATAATCCCAAGCG-3'), and Rps9 (forward primer: 5'-GACCAGGAGCTAAAGTTGA

TTGGA-3'; reverse primer: 5'-TCTTGGCCAGGGTAAACTTGA-3'; probe: 5'-AAACCTCACGTTTG TTCCGGAGTCCATACT-3').

## Spontaneous alternation and exploratory behavior in the Y-maze

Mice were habituated by an adapted tunnel-handling protocol (*Hurst and West, 2010*), which minimizes the stress associated with picking up and moving mice. In brief, all animals were exposed to a Plexiglas tunnel for 3 days before the first experimental session. They were habituated by gently picking them up repeatedly for 30 s with the tunnel and allowing them to recover for 1 min in between. This protocol was repeated on 2 days, twice per day at random time points, during the week prior to the experiments.

To test working memory, continuous spontaneous alternation behavior (SAB) was assessed in the Y-Maze (three arms of equal size symmetrically placed at 120°; inner length: 36.5 cm, width: 6 cm, height: 15 cm, color: grey) described in detail elsewhere (*Hughes, 2004*; *Ramanathan et al., 2010*). Each mouse was picked up with the tunnel, released to the center of the maze and allowed to explore freely for 7 min, before being returned with the tunnel to the home cage. The maze was surrounded by black walls to avoid any extra-maze cues and cleaned with 1% acetic acid between mice. Testing was performed from ZT 22.5–24 in the dark phase, the main activity phase of mice, under indirect dim light (15 lux). Each animal was tested three times: run one at the end of the first recovery day after sleep deprivation (i.e. after 16.5 hr of recovery), run two followed 24 hr later, whereas run three was performed 1 week after sleep deprivation (6 days after run two). Number of control animals: 6 WT, 8 HT and 6 KO mice; numbers of sleep deprived animals: 7 WT, 9 HT and 10 KO mice. Mouse behavior in the maze was video-recorded from above and arm entry sequences assigned visually. Spontaneous alternations [%] were defined as successful entry sequences (i.e. all three arms visited in three consecutive entries) relative to possible successful entry sequences (i.e. total number of entries minus two). An arm was considered to be entered, when the mouse had placed all four paws within the walls of the arm (*Hughes, 2004*). Mice were tested in four batches. Two parameters were analyzed from the Y-maze data: 'Alternation score' was used as a measure of working memory, whereas the 'number of total arm entries' was used as a measure for exploratory behavior.

## Statistical analyses

Statistical analyses were performed with SAS 9.1.3 software (SAS Institute, Cary, NC) for human data and in R-project 3.1.2 software (www.r-project.org) for mice analysis. To examine associations between mGluR5 availability and EEG markers of sleep need, Spearman rank correlations were calculated. Pearson's product moment correlation analyses revealed very similar results, and are not reported here. To correct for scan order in the MR spectroscopy imaging, two-way repeated measure mixed model ANOVAs were performed with factors 'scan-order' (first vs second) and condition (sleep control vs sleep deprived). For remaining effects of sleep-deprivation, two tailed paired Student's t-tests were performed. Before statistical testing, variables were tested for normality. If the distribution significantly differed from a Gaussian distribution, appropriate transformations were applied for statistical analyses.

To correct for multiple comparisons when region-specific associations between mGluR5 availability and EEG <1 Hz activity were examined, the significance level was set to $\alpha$ <0.00278 (Bonferroni correction: $\alpha$ = 0.05/18 [2 conditions x 9 regions]). A significant contribution of regional mGluR5 availability to EEG <1 Hz activity was concluded only when the corrected $\alpha$ level was reached in both, baseline and sleep deprivation conditions (*Table 1*).

The behavioral data in mice were statistically analyzed by ANOVA with factors 'genotype' x 'condition' x 'run' x 'batch', and followed by appropriate post-hoc t-tests if significance was found for the respective factors or interactions.

## Acknowledgements

This work was supported by the Swiss National Science Foundation grants # 135414 and # 163439 (to HPL) and # 146615 (to MT), the Clinical Research Priority Program of the University of Zürich '*Sleep and Health*', and the National Center for Competence in Research '*Neural Plasticity and Repair*'. We thank Dr. R Wehrle, Dr. R Dürr, Dr. V Bachmann, Th Berthold, Dr. C Klein, C

Siegenthaler, S Röthlisberger, M Röthlisberger, C Schneider, C Laengle, S Jimenez, Y Emmenegger, and R Abbas for their help with data collection and analyses.

## Additional information

### Funding

| Funder | Grant reference number | Author |
| --- | --- | --- |
| Swiss National Science Foundation | 320030_135414 | Hans-Peter Landolt |
| Universität Zürich | Sleep and Health | Hans-Peter Landolt |
| NCCR Neural Plasticity and Repair | | Erich Seifritz Hans-Peter Landolt |

The funders had no role in study design, data collection and interpretation, or the decision to submit the work for publication.

### Author contributions

Sebastian C Holst, Data curation, Software, Formal analysis, Supervision, Funding acquisition, Investigation, Visualization, Writing—original draft, Writing—review and editing; Alexandra Sousek, Data curation, Formal analysis, Investigation, Visualization, Writing—original draft, Writing—review and editing; Katharina Hefti, Data curation, Investigation; Sohrab Saberi-Moghadam, Formal analysis; Alfred Buck, Simon M Ametamey, Resources, Methodology, Project administration; Milan Scheidegger, Investigation, Methodology, Writing—review and editing; Paul Franken, Resources, Investigation, Writing—review and editing; Anke Henning, Software, Methodology; Erich Seifritz, Investigation, Project administration, Writing—review and editing; Mehdi Tafti, Resources, Supervision, Project administration, Writing—review and editing; Hans-Peter Landolt, Conceptualization, Resources, Data curation, Supervision, Funding acquisition, Writing—original draft, Project administration, Writing—review and editing

### Author ORCIDs

Sebastian C Holst (iD) https://orcid.org/0000-0003-3657-4535
Paul Franken (iD) http://orcid.org/0000-0002-2500-2921
Mehdi Tafti (iD) https://orcid.org/0000-0002-6997-3914
Hans-Peter Landolt (iD) http://orcid.org/0000-0003-0887-9403

### Ethics

Human subjects: All experimental procedures were conducted in accordance with the declaration of Helsinki (1964) and approved by the cantonal (ethics committee for research on human subjects of the canon of Zurich [Reference Nr. EK-Nr. 786] and Swiss federal authorities for research on human (Swiss Federal Institute of Public Health, Reference Nr. 464-0002-6/08.005701) subjects.
Animal experimentation: All animal experiments were carried out in accordance with the regulations of the Swiss Federal and State of Vaud Veterinary Offices (No. 2699.0).

### Decision letter and Author response

Decision letter https://doi.org/10.7554/eLife.28751.023
Author response https://doi.org/10.7554/eLife.28751.024

## Additional files

### Supplementary files

• Transparent reporting form
DOI: https://doi.org/10.7554/eLife.28751.022

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
