## [Decision Letter]

Thank you for submitting your article "Cerebral mGluR5 Availability Contributes to Elevated Sleep Need and Behavioral Adjustment after Sleep Deprivation" for consideration by *eLife*. Your article has been reviewed by three peer reviewers, one of whom is a member of our Board of Reviewing Editor and the evaluation has been overseen by Eve Marder as the Senior Editor. The following individuals involved in review of your submission have agreed to reveal their identity: Michael Lazarus (Reviewer #2); Anita Lüthi (Reviewer #3).

The reviewers have discussed the reviews with one another and the Reviewing Editor has drafted this decision to help you prepare a revised submission.

The present work is an interesting study combining human and mouse experimentation that lends further support to a molecular element in homeostatic sleep regulation: the metabotropic glutamate receptor 5 and associated signaling pathways. Direct measurements of mGluR5 availability and associated signaling in humans through PET and subsequent MRI are combined with spectral and behavioral assessments of sleep deprivation and early baseline sleep. Sleep-wake behavior in mGluR5-deficient mice unravels additional complex roles of this receptor in sleep-wake regulation and behavioral adaptation to sleep pressure.

Summary:

In this manuscript, the authors found that the availability of mGluR5 in human brain is positively correlated with EEG δ activity of recovery sleep. Consistent with the human results, they also found that mice without mGluR5 gene showed retarded increase of EEG δ power in the dark phase and after sleep deprivation. Consequently, the mGluR5 knockout mice showed greatly reduced rebound sleep after sleep deprivation. This study provides strong evidence for the function of mGluR5 in regulating mammalian sleep homeostasis, suggesting that mGluR5 is indispensable for the accumulation of sleep need after sustained wakefulness.

A sleep phenotype of mGluR5-/- mice has been reported by another group in 2015 (Ahnaou et al., 2015). The phenotype of mGluR5-/- mice described by Ahnaou et al. is somewhat different than reported here. In their study, the mGluR5-/- mice showed no increased wakefulness and no reduced NREM sleep after sleep deprivation. So the authors need to present and discuss these apparently contradictory results.

Essential revisions:

1) The author's statements in subsection “Sleep deprivation-induced changes in brain metabolites reflect downstream markers of mGluR5 activation “("Collectively, mGluR5 contribute to sleep regulation by affecting downstream mechanisms of mGluR5-mediated protein phosphorylation and enhanced N-methyl-D-aspartate (NMDA) receptor-mediated signaling") and subsection “mGluR5 availability predicts behavioral and EEG markers of sleep need in humans” ("[…]but represents a molecular marker of elevated sleep need in response to sleep loss in humans.") are overreaching based on evidence provided by the human studies. One important question that the authors should address is mGluR5 availability or expression levels in sleep-deprived mice (wild-type and KO mice) as compared to control mice. This may provide a link between human imaging studies and mouse behavior to support their hypothesis that mGluR5 is important for sleep need.

2) The authors state in subsection “mGluR5 availability predicts behavioral and EEG markers of sleep need in humans” that "A similar relationship was also observed for the entire 0.5-4.5 Hz band (yet not any other frequency band)[…] ". It would be informative if the data for the other frequencies are shown.

3) The authors argue that altered EEG δ power dynamics are suggesting "a deficient build-up of homeostatic sleep need in wakefulness". It is however possible that changes in δ power in the KO mice is independent from the homeostatic sleep need, because there are no differences in baseline sleep between the genotypes. Moreover, the statement "… demonstrating a severely disturbed sleep homeostatic response to prolonged wakefulness." (subsection “Important role for mGluR5 in homeostatic response to sleep deprivation”) is at odds with Figure 5—figure supplement 1, in which the KO mice actually show a higher NREM sleep amount after sleep deprivation in the remaining light phase, indicating a homeostatic sleep response exists which may just be stronger than in wild-type mice during this period (i.e. the dissipation of sleep need may be enhanced).

4) In the Discussion section, the authors suggest that sleep loss causes an increase in glycine levels. What could be the reason for this rise in glycine by sleep deprivation? Without a reasonable explanation, it is possible that this is unrelated to the mGluR5/NMDA signalling system.

5) The statement "Thus, increased mGluR5 signaling during prolonged periods of spontaneous or enforced wakefulness may aid or facilitate sustained wakefulness" (subsection “Lack of mGluR5 compromises adjustment to novel environment after sleep deprivation”) is inconsistent with the fact that mGluR5 KO mice have a lower amount of recovery sleep and SWA. The authors need to define more carefully the role of mGluR5 in sleep (or wake) regulation. What does a prolonged period of spontaneous wakefulness look like?

6) One asset of this work is the combination of studies on human and mouse. Could this be pushed even further? The human studies include recovery sleep after sleep deprivation. Were there interindividual variabilities in this recovery sleep that correlated with the availability of mGluR5? Can anything be said about the activity profile in the day after recovery sleep?

7) The interindividual variability in mGluR5 availability and its correlation with baseline sleep properties a remarkable observation that can help explain basic differences in sleep need. Have any assessments been done with the subjects in this direction, e.g. via questionnaires? Were there any differences in mood-related behaviors?

8) The interindividual variability should be better emphasized in the paper. For example, in Figure 2, it is not clear which points in Figure 2 relate to the points in Figure 2. From Figure 2, it can be guessed that there is a subgroup of subjects that show particularly large increases in mGluR5 availability. Were low mGluR5-expressers particularly vulnerable to SD in terms of mGluR5 changes?

9) Why were maximal availability increases after SD not higher than the maximal availability values in baseline?

10) The use of frequency bands in the power spectra of NREM sleep needs to be better described. In Figure 2, the full δ (0.5-4 Hz) and the high δ (2-4 Hz) bands are analyzed. In Figure 3, it is the slow wave band (< 1 Hz) that is emphasized. Although it is very interesting to note that in particular power in the high δ band correlates with mGluR5 increases, this needs to be clearly motivated. Same for Figure 3.

11) Homer 1a is also upregulated with SD, and Homer1a uncouples mGluR5 from its intracellular signaling partners. However, myo-inositol levels are decreased in a manner that correlate with decreases in mGluR5 levels. Are decreased myo-inositol levels also correlated with enhanced low-frequency power levels in the spectrogram? Is this to conclude that augmented mGluR5 activation overcomes Homer1a upregulation? The results seemed to be pointing to this somewhat paradoxical situation, and should be further discussed.

12) Three subjects were excluded from PET imaging, so why n =26 in subsection “mGluR5 availability predicts behavioral and EEG markers of sleep need in humans”?

13) In a supplementary figure, can example metabolites be presented to better illustrate the criteria for exclusion/inclusion as described in subsection “Sleep deprivation-induced changes in brain metabolites reflect downstream markers of mGluR5 activation”?

---

## [Author Response]

Summary:In this manuscript, the authors found that the availability of mGluR5 in human brain is positively correlated with EEG δ activity of recovery sleep. Consistent with the human results, they also found that mice without mGluR5 gene showed retarded increase of EEG δ power in the dark phase and after sleep deprivation. Consequently, the mGluR5 knockout mice showed greatly reduced rebound sleep after sleep deprivation. This study provides strong evidence for the function of mGluR5 in regulating mammalian sleep homeostasis, suggesting that mGluR5 is indispensable for the accumulation of sleep need after sustained wakefulness.A sleep phenotype of mGluR5-/- mice has been reported by another group in 2015 (Ahnaou et al., 2015). The phenotype of mGluR5-/- mice described by Ahnaou et al. is somewhat different than reported here. In their study, the mGluR5-/- mice showed no increased wakefulness and no reduced NREM sleep after sleep deprivation. So the authors need to present and discuss these apparently contradictory results.

Consistent with our results, Ahnaou et al. (2015) reported reduced EEG slow-wave activity (SWA) and sleep drive in mGluR5 knock-out (KO) mice after sleep deprivation. However, the profound changes in sleep amounts and EEG δ power in the dark period after sleep deprivation were not found by the previous authors. The discrepancy may be due to several methodological differences between the studies, including different durations and techniques of sleep deprivation (automated procedure vs. gentle handling), different EEG-EMG recording devices (implants vs. cables), etc. Most importantly, Ahnaou et al. employed an automated sleep deprivation technique with forced locomotion during 8 hours, while we have used the well-established gentle handling sleep deprivation for 6 hours. Prolonged forced locomotion induces a stereotypic behavior different from normal wakefulness, which decreases cortical activity and the rebound in subsequent sleep δ activity (Fisher et al., 2016; Franken et al., 2001). In the revised manuscript, we write (subsection “Genetic loss-of-function of mGluR5 in mice causes dysregulation of sleep-wake patterns and sleep EEG δ activity”): “In accordance with our findings, a diminished rebound in NREM sleep EEG δ power after prolonged wakefulness in mGluR5 KO animals was reported previously (Ahnaou et al., 2015). Nevertheless, the profound changes in sleep amounts and the dynamics of EEG δ power observed here were not found in the previous study. The discrepancy between the studies may be due to several methodological differences, in particular the use of an automated sleep deprivation technique by the other authors which induces forced locomotion. Forced locomotion causes stereotypic behavior unequal to normal wakefulness and the dissipation of sleep pressure (Fisher et al., 2016).”

Essential revisions:1) The author's statements in subsection “Sleep deprivation-induced changes in brain metabolites reflect downstream markers of mGluR5 activation “("Collectively, mGluR5 contribute to sleep regulation by affecting downstream mechanisms of mGluR5-mediated protein phosphorylation and enhanced N-methyl-D-aspartate (NMDA) receptor-mediated signaling") and subsection “mGluR5 availability predicts behavioral and EEG markers of sleep need in humans” ("…, but represents a molecular marker of elevated sleep need in response to sleep loss in humans.") are overreaching based on evidence provided by the human studies.

We agree with the reviewers that the above statements are overreaching based on the data presented. However, we have been more careful in interpreting our results than implied above. In subsection “mGluR5 availability predicts behavioral and EEG markers of sleep need in humans“, we wrote in the original version of the manuscript: “These results provide the first puzzle piece of evidence for the hypothesis that functional mGluR5 availability not only correlates with absolute low-frequency EEG power, but represents a molecular marker of elevated sleep need in response to sleep loss in humans.” Furthermore, in subsection “Sleep deprivation-induced changes in brain metabolites may reflect downstream markers of mGluR5 activation”, we were careful in writing: “Collectively, these findings suggest that mGluR5 contribute to sleep regulation by affecting downstream mechanisms of mGluR5-mediated protein phosphorylation and enhanced N-methyl-D-aspartate (NMDA) receptor-mediated signaling.” To be even more careful, we added “may suggest” in the revised version of the manuscript.

One important question that the authors should address is mGluR5 availability or expression levels in sleep-deprived mice (wild-type and KO mice) as compared to control mice. This may provide a link between human imaging studies and mouse behavior to support their hypothesis that mGluR5 is important for sleep need.

We thank the reviewers’ for the suggestion. We performed qPCR analyses of mGluR5 mRNA expression extracted from cortex, hippocampus and striatum in mGluR5 wild type (WT), heterozygous (HT) and KO mice and found no consistent effect of sleep deprivation on mGluR5 mRNA expression. We report these experiments in new Figure 5.

Other authors recently also failed to find differences in mGluR5 protein expression between wakefulness and sleep (Diering et al., 2017, Figure S1). Therefore, the sleep deprivation-induced changes in mGluR5 availability found in our human experiment are likely to indicate a functional synaptic change due to receptor trafficking rather than a change in overall mRNA or protein levels. To replicate the human findings in mice, autoradiographic quantification could be performed in future studies. We write in the revised version of the manuscript (subsection “Lack of mGluR5 in mice interferes with sleep rebound after sleep deprivation “): “Furthermore, we performed qPCR analyses of mGluR5 mRNA expression extracted from cortex, hippocampus and striatum in mGluR5 WT, HT and KO mice after sleep control and sleep deprivation. While mGluR5 expression importantly varied according to allele ‘dose’, we did not observe a consistent effect of sleep loss on mGluR5 mRNA expression in WT and HT mice (Figure 5). Similarly, other authors recently failed to find differences in mGluR5 protein expression between wakefulness and sleep (Diering et al., 2017). Therefore, the sleep deprivation-induced changes in mGluR5 availability we found in our human experiment are likely representing a functional synaptic change due to receptor trafficking rather than changes in overall mRNA or protein levels.”

2) The authors state in subsection “mGluR5 availability predicts behavioral and EEG markers of sleep need in humans” that "A similar relationship was also observed for the entire 0.5-4.5 Hz band (yet not any other frequency band)…, ". It would be informative if the data for the other frequencies are shown.

We thank the reviewers for this suggestion. We agree that the higher frequency bands are interesting and potentially valuable to present. They were omitted in the original manuscript because the statistical analyses revealed no significant correlations with regional mGluR5 availability. We now present a supplement to Figure 3, including the correlation analyses for five predefined frequency ranges between 0.5 and 20 Hz. We emphasize in the revised manuscript that with the exception of the δ band, the statistical results do not withstand correction for multiple comparison.

3) The authors argue that altered EEG δ power dynamics are suggesting "a deficient build-up of homeostatic sleep need in wakefulness". It is however possible that changes in δ power in the KO mice is independent from the homeostatic sleep need, because there are no differences in baseline sleep between the genotypes. Moreover, the statement "… demonstrating a severely disturbed sleep homeostatic response to prolonged wakefulness." (subsection “Important role for mGluR5 in homeostatic response to sleep deprivation”) is at odds with Figure 5—figure supplement 1, in which the KO mice actually show a higher NREM sleep amount after sleep deprivation in the remaining light phase, indicating a homeostatic sleep response exists which may just be stronger than in wild-type mice during this period (i.e. the dissipation of sleep need may be enhanced).

Baseline differences in time-spent-asleep cannot be taken as evidence of a different homeostatic ‘need’. That would imply that all sleep expressed under undisturbed baseline conditions is homeostatically defended, which we know is not the case and we also know that factors other than homeostatic drive can influence undisturbed time-spent asleep. Moreover, would less sleep time signal a higher sleep pressure as a result of sleeping less or a reduced need for sleep?

The sleep-wake dependent changes in δ power are widely used as markers of momentary sleep ‘need’ because the longer one is awake, the higher δ power in subsequent NREM sleep will be. Thus, the lack of a genotype difference in the amount and distribution of sleep during baseline in the presence of a marked blunting of the build-up of δ power in the dark or active period shows that the quantitative relationship between time-spent-awake and δ power is severely altered in KO mice. If the reviewers can agree that δ power is a marker of a homeostatic process then one is allowed to interpret this finding to indicate that per unit time spent awake sleep need increases less when mGluR5 is missing. The observations made after an enforced period of wakefulness (i.e., sleep deprivation) confirms this interpretation. By contrast, δ power normally decays during sleep (i.e., during the light periods of baseline and recovery).

4) In the Discussion section, the authors suggest that sleep loss causes an increase in glycine levels. What could be the reason for this rise in glycine by sleep deprivation? Without a reasonable explanation, it is possible that this is unrelated to the mGluR5/NMDA signalling system.We agree with the reviewers that the spectroscopic data of our study need to be interpreted with caution, yet they open up possibly promising new avenues for future research. We added a cautionary note in the Discussion section: “In view of the diverse effects of glycine on wakefulness and sleep, the elusive mechanisms underlying its rise by sleep deprivation, as well as glycine’s varying functions in different neuronal circuits (Giber et al., 2015; Zeilhofer, 2005), the present data need to be interpreted with caution. Nevertheless, the spectroscopic findings may indicate that the mGluR5-associated Homer1a-IP_3_ and glycine-NMDA receptor pathways importantly contribute to the molecular machinery that regulates sleep-wake homeostasis in humans and open up promising new avenues for future research.”5) The statement "Thus, increased mGluR5 signaling during prolonged periods of spontaneous or enforced wakefulness may aid or facilitate sustained wakefulness" (subsection “Lack of mGluR5 compromises adjustment to novel environment after sleep deprivation”) is inconsistent with the fact that mGluR5 KO mice have a lower amount of recovery sleep and SWA. The authors need to define more carefully the role of mGluR5 in sleep (or wake) regulation. What does a prolonged period of spontaneous wakefulness look like?

We agree with the reviewers that this part of the Discussion section was partially unprecise. We revised the respective paragraph: “Thus, sleep-wake dependent changes in mGluR5 signaling may aid or facilitate sustained wakefulness and the proper homeostatic build-up of sleep propensity during wakefulness as reflected in EEG δ power in NREM sleep. At the same time, mGluR5-dependent mechanisms may promote and maintain deep sleep rich of slow waves.”

6) One asset of this work is the combination of studies on human and mouse. Could this be pushed even further? The human studies include recovery sleep after sleep deprivation. Were there interindividual variabilities in this recovery sleep that correlated with the availability of mGluR5? Can anything be said about the activity profile in the day after recovery sleep?

We thank the reviewers for their detailed interest in these results.

To further investigate the relationships between recovery sleep after sleep deprivation and inter-individual variation in mGluR5 availability, we created additional supplements to Figure 2. First, we subdivided the data by median split into study participants with low and high change in global mGluR5 availability after sleep deprivation, and compared the effect of prolonged wakefulness on EEG δ and slow-oscillation activity. These analyses revealed that the group with a minor change in mGluR5 availability also showed a significantly reduced increase in δ and < 1 Hz activity when compared to the group with a more pronounced increase in mGluR5. By contrast, neither theta, α, σ (p = 0.06) nor β activity were differently changed by sleep deprivation in the two groups.

7) The interindividual variability in mGluR5 availability and its correlation with baseline sleep properties a remarkable observation that can help explain basic differences in sleep need. Have any assessments been done with the subjects in this direction, e.g. via questionnaires? Were there any differences in mood-related behaviors?

We thank the reviewers for this interesting question.

The associations between sleep deprivation-induced changes in subjective measures and mGluR5 availability were the focus of our previous publication (Hefti et al. 2013). The current manuscript focuses on the associations between objective markers of elevated sleep need after prolonged wakefulness and mGluR5 availability.

Given the interesting nature of the above question, however, we performed additional analyses to shed some light on possible associations between mGluR5 availability in baseline and mood. Mood assessments with the 24-item Profile of Mood States (POMS) were performed at 3-hour intervals during extended waking and before each imaging session. Investigating POMS scores collected immediately before the baseline PET scans, and the average scores for the six assessments across the baseline day, did not reveal significant correlations with global mGluR5 availability in baseline (p_all_ > 0.1). Variables tested included the total mood score and the subcategories depression, fatigue, vigor and irritability. Given the negative statistical tests results, these new analyses are not included in the manuscript. It needs to be kept in mind, however, that a homogenous group of carefully screened healthy volunteers was studied.

Whether mGluR5 availability in specific brain regions could be associated with mood measures (such as found in depressed patients; Deschwanden et al., 2011) or whether other mood questionnaires or test scores may be more sensitive than the POMS to capture interindividual variation in mGluR5 availability remain possible subjects for future research.

8) The interindividual variability should be better emphasized in the paper. For example, in Figure 2, it is not clear which points in Figure 2 relate to the points in Figure 2. From Figure 2, it can be guessed that there is a subgroup of subjects that show particularly large increases in mGluR5 availability. Were low mGluR5-expressers particularly vulnerable to SD in terms of mGluR5 changes?

We agree with the reviewers that the inter-subject variability in mGluR5 availability and individual responses to sleep deprivation can be better illustrated. To show each individual’s response, we created an additional supplement to Figure 2. In addition, we split the subjects based on either low (n = 11) or high (n = 12) global mGluR5 availability at baseline (median split) and found that those study participants with a lower baseline expression of mGluR5 exhibit a significantly larger increase by sleep loss than those participants with a higher baseline mGluR5 availability.

9) Why were maximal availability increases after SD not higher than the maximal availability values in baseline?

The global change in mGluR5 availability by sleep deprivation equals roughly 5% , which makes it difficult to capture with the naked eye. As illustrated above (reviewer item # 8), the participants with the highest mGluR5 availability in baseline typically showed a minor increase, or even a slight decrease, in mGluR5 availability after sleep deprivation. Interestingly, a similar bi-directional relationship was recently reported for the effects of sleep deprivation on markers of associative synaptic plasticity in the human cortex (Kuhn et al., 2016). These authors suggested that their findings could be explained by the Bienenstock-Cooper-Munro theory of bi-directional synaptic plasticity, stating that the threshold for LTP (long-term potentiation/LTD (long-term depression) induction is adjusted to the level of prior synaptic activity (Bienenstock et al., 1982). Our data may offer the fascinating possibility that sleep deprivation-induced changes in mGluR5 availability provide a molecular substrate contributing to the observed effects of sleep loss on LTP- and LTD-like plasticity. Alternatively, high baseline mGluR5 availability may be difficult to increase further by increased time awake and the data may reflect a ceiling effect. Further studies are necessary to investigate the different hypotheses.

10) The use of frequency bands in the power spectra of NREM sleep needs to be better described. In Figure 2, the full δ (0.5-4 Hz) and the high δ (2-4 Hz) bands are analyzed. In Figure 3, it is the slow wave band (< 1 Hz) that is emphasized. Although it is very interesting to note that in particular power in the high δ band correlates with mGluR5 increases, this needs to be clearly motivated. Same for Figure 3.

We agree with the reviewers that the presentation of the different frequency bands could have been better explained in the manuscript.

The study’s primary aim was to address the hypothesis that mGluR5-related mechanisms are associated with sleep homeostasis. For that reason, we focused on the EEG δ range below 5 Hz. In the initial analysis steps, we only considered frequency bands relevant if multiple adjacent frequency bins showed a significant (p < 0.05) correlation between mGluR5 availability and consecutive 0.25-Hz bins in the EEG δ range at baseline and sleep deprivation. This analysis revealed highly significant correlations in the bins < 1 Hz. Interestingly, the 0.5-4.5 Hz band remained significant yet generally with lower correlation coefficients than the < 1 Hz bins. When examining the relative correlations, i.e., how EEG activity and mGluR5 availability changed with sleep deprivation, the high δ range > 2 Hz was significant.

To further illustrate the frequency specificity of these associations, the figure below shows the Spearman rank correlation coefficients between global mGluR5 availability and EEG power between 0-20 Hz at baseline, sleep deprivation and the change caused by sleep loss. The green lines indicate r-values that are above (or below) the a = 0.05 significance threshold. The Figure 2—figure supplement 2 illustrates that the 2-4 Hz range is relevant for the change after sleep deprivation.

11) Homer 1a is also upregulated with SD, and Homer1a uncouples mGluR5 from its intracellular signaling partners. However, myo-inositol levels are decreased in a manner that correlate with decreases in mGluR5 levels. Are decreased myo-inositol levels also correlated with enhanced low-frequency power levels in the spectrogram? Is this to conclude that augmented mGluR5 activation overcomes Homer1a upregulation? The results seemed to be pointing to this somewhat paradoxical situation, and should be further discussed.

This is an interesting question, which is difficult to answer at present.

As discussed in the manuscript, our results are in line with published data in rats showing that myo-inositol is inversely correlated with neuronal activity (Xu et al., 2005). In addition, our findings are consistent with a very recent publication (available on-line) in male adolescents reporting that frontal cortex myo-inositol levels correlated negatively with subjective sleepiness and positively with total sleep time (Urrila et al., in press). We did not measure Homer1a expression in humans before and after sleep deprivation, nor, to our knowledge, has this been quantified in any other published study. Future work is warranted, to simultaneously quantify mGluR5 and Homer1a expression as a function of time awake.

When correlating myo-inositol concentrations with EEG power in the low frequency range, we found marginally significant associations with < 1-Hz activity (baseline: r = 0.47, p < 0.06; sleep deprivation: r = 0.60, p < 0.02). By contrast, the sleep deprivation-induced change in myo-inositol did not correlate significantly with the relative changes in single 0.25-Hz bins and the entire band within the 0.5-4.5 Hz range after sleep deprivation.

12) Three subjects were excluded from PET imaging, so why n =26 in subsection “mGluR5 availability predicts behavioral and EEG markers of sleep need in humans”?

We thank the reviewers for pointing out this mistake. It has been corrected in the revised manuscript (subsection “mGluR5 availability predicts behavioral and EEG markers of sleep need in humans”).

13) In a supplementary figure, can example metabolites be presented to better illustrate the criteria for exclusion/inclusion as described in subsection “Sleep deprivation-induced changes in brain metabolites reflect downstream markers of mGluR5 activation”?

To better illustrate the criteria for the selection of relevant metabolites, the statistical test results are summarized in the Table below. Only those metabolites that showed a significant alteration by sleep deprivation and, simultaneously, significant correlation with global mGluR5 availability in baseline and sleep deprivation conditions or with the sleep deprivation-induced increase in global mGluR5 availability, were considered relevant.

**Gly****mi****Cho****Cr****Gln****Glu****GABA****NAA**Δ sleep deprivation (factor ‘condition’)**F_1,14_ = 6.27, p <.03****F_1,14_ = 4.95, p <.05**F_1,14_ = 1.61, p >.22F_1,14_ = 1.27, p >.27F_1,14_ = 0.17, p >.68F_1,14_ = 0.28, p >.60F_1,14_ = 0.21, p >.65F_1,14_ = 0.15, p >.82Correlation withmGluR5 _Baseline_r =.34, p >.17**r =.54, p <.03**r =.39, p >.11**r =.54, p <.03**r =.41, p >.10r =.20, p >.43r =.01, p >.95r =.17, p >.49mGluR5 _SD_r =.45, p <.10**r =.56, p <.04****r =.70, p <.005****r =.67, p <.008**r = -.42, p >.13r = -.23, p >.41r =.09, p >.80r =.30, p >.28mGluR5 _Δ SD_**r =.53, p <.05**r= -.22 p >.43r = -.01, p >.94r =.11, p >.69r =.17, p >.54r = -.02, p >.93r = -.28, p >.53r =.49, p <.08

**Legend:** Change with sleep deprivation refers to a two-way repeated measure mixed model with the factors ‘condition’ and ‘scan-order’. Metabolites were normalized to approximate a Gaussian distribution when appropriate. Pearson’ correlation coefficients were computed with global DV_norm_ of mGluR5. Gly: Glycine; mi: myo-inositol; Cho: choline; Cr: total creatine and phosphocreatine; Gln: glutamine; Glu: glutamate; GABA: γ-Aminobutyric acid; NAA: N-acetylaspartate. Spearman rank correlation coefficient and p-values are indicated. SD = sleep deprivation.